# Cadherin 11 Inhibition Downregulates β-catenin, Deactivates the Canonical WNT Signalling Pathway and Suppresses the Cancer Stem Cell-Like Phenotype of Triple Negative Breast Cancer

**DOI:** 10.3390/jcm8020148

**Published:** 2019-01-27

**Authors:** Pamungkas Bagus Satriyo, Oluwaseun Adebayo Bamodu, Jia-Hong Chen, Teguh Aryandono, Sofia Mubarika Haryana, Chi-Tai Yeh, Tsu-Yi Chao

**Affiliations:** 1International Ph.D. Program in Medicine, College of Medicine, Taipei Medical University, Taipei City 11031, Taiwan; pbagus57@yahoo.com; 2Doctorate Program of Medical and Health Science, Faculty of Medicine Public Health and Nursing, Universitas Gadjah Mada, Yogyakarta 55281, Indonesia; 3Department of Hematology & Oncology, Taipei Medical University-Shuang Ho Hospital, New Taipei City 23561, Taiwan; dr_bamodu@yahoo.com; 4Department of Medical Research & Education, Taipei Medical University-Shuang Ho Hospital, New Taipei City 23561, Taiwan; 5Graduate Institute of Clinical Medicine, College of Medicine, Taipei Medical University, Taipei City 11031, Taiwan; ndmc_tw.tw@yahoo.com.tw; 6Division of Medical Oncology and Hematology, Tri-Service General Hospital, National Defense Medical Centre, Taipei 11409, Taiwan; 7Department of Surgery, Faculty of Medicine Public Health and Nursing, Universitas Gadjah Mada, Yogyakarta 55281, Indonesia; teguharyandono@yahoo.com; 8Department of Histology and Cellular Biology, Faculty of Medicine Public Health and Nursing, Universitas Gadjah Mada, Yogyakarta 55281, Indonesia; sofia.mubarika@gmail.com

**Keywords:** Cadherin 11, WNT signaling, β-catenin, cancer stem cells, TNBC

## Abstract

Background: Cancer stem cells (CSCs) promote tumor progression and distant metastasis in breast cancer. Cadherin 11 (CDH11) is overexpressed in invasive breast cancer cells and implicated in distant bone metastases in several cancers. The WNT signalling pathway regulates CSC activity. Growing evidence suggest that cadherins play critical roles in WNT signalling pathway. However, CDH11 role in canonical WNT signalling and CSCs in breast cancer is poorly understood. Methods: We investigated the functional association between CDH11 and WNT signalling pathway in triple negative breast cancer (TNBC), by analyzing their expression profile in the TCGA Breast Cancer (BRCA) cohort and immunohistochemical (IHC) staining of TNBC samples. Results: We observed a significant correlation between high CDH11 expression and poor prognosis in the basal and TNBC subtypes. Also, CDH11 expression positively correlated with β-catenin, wingless type MMTV integration site (WNT)2, and transcription factor (TCF)12 expression. IHC results showed CDH11 and β-catenin expression significantly correlated in TNBC patients (*p* < 0.05). We also showed that siRNA-mediated loss-of-CDH11 (siCDH11) function decreases β-catenin, Met, c-Myc, and matrix metalloproteinase (MMP)7 expression level in MDA-MB-231 and Hs578t. Interestingly, immunofluorescence staining showed that siCDH11 reduced β-catenin nuclear localization and attenuated TNBC cell migration, invasion and tumorsphere-formation. Of translational relevance, siCDH11 exhibited significant anticancer efficacy in murine tumor xenograft models, as demonstrated by reduced tumor-size, inhibited tumor growth and longer survival time. Conclusions: Our findings indicate that by modulating β-catenin, CDH11 regulates the canonical WNT signalling pathway. CDH11 inhibition suppresses the CSC-like phenotypes and tumor growth of TNBC cells and represents a novel therapeutic approach in TNBC treatment.

## 1. Introduction

Triple negative breast cancer (TNBC) constitutes 10−20% of all breast cancer cases and exhibits more aggressive traits and worse patient prognosis than hormone receptor- and HER2- positive breast cancer [1]. To date, TNBC lacks US-FDA-approved target therapy [1,2]. Chemotherapy remains the primary treatment of choice. Despite a documented record of TNBC sensitive to chemotherapy, only 30% of such cases show pathologic complete responses, while the less- or non- sensitive patients exhibit a high probability of developing distant metastases and relapses [2]. Metastatic breast cancer is currently an incurable disease, with palliative treatment being the therapy of last resort to alleviate pain, enhance quality of life and help patients live longer [3]. The discovery and development of novel efficacious targeted therapy to treat TNBC and prevent metastatic breast disease are important issues. 

Since the discovery of CSCs in solid tumors, namely, breast cancer in 2003 [4], there is growing evidence of CSCs existence or presence in other solid tumors [5]. The CSCs possess self-renewal capacity and differentiate into heterogeneous tumor cells. They have been implicated in metastatic dissemination of cancerous cells in many cancer models including breast cancer and are shown to be more resistant to chemotherapy and lead to cancer relapse [6]. Targeting these cells in cancer treatment may be more beneficial for TNBC patients.

The WNT signalling pathway is a principal regulator of self-renewal and helps maintain the undifferentiated state of normal mesenchymal stem cells and CSCs [7,8]. The presence of the WNT ligand has been shown to promote β-catenin translocation to the nucleus and activate target gene of WNT signalling pathway [9]. Knocking down β-catenin reduced stem cell-like cell population and their chemoresistance to doxorubicin; the β-catenin-silenced TNBC cell lines generated smaller tumors than control cells when inoculated into the mammary fat pad of tumor xenograft mice models [10]. In the membrane, β-catenin binds to different cadherins in the adhesion junction complex and become stabilized; however, β-catenin in the cytoplasm is degraded by the APC-Axin-GSK3β complex in the absence of WNT ligands [11]. In fact, cadherins inhibit activation of the canonical WNT signalling pathway by keeping β-catenin in the membrane. β-catenin translocation to the nucleus is prevented in the presence of WNT ligands [12]. During epithelial-to-mesenchymal transition (EMT), cadherins act as a pool of calcium-dependent adhesion-competent β-catenin, which is a principal mediator of canonical WNT signalling activation, thus, depleting cadherins reduces β-catenin in the membrane and suppresses WNT signalling activation despite the presence of β-catenin in the cytoplasm [13]. Cadherin 11, a type II cadherin and mesenchymal protein marker, is overexpressed in the invasive breast cancers [14]. In prostate and renal cancers, CDH11 plays an important role in distant bone metastasis. Expression of CDH11 in metastatic bone tumors is higher than in primary tumor sites [15,16]. The brain, lungs, and bones are the most common sites of distant metastasis in TNBC patients [17,18]. Interestingly, these organs have higher expression of CDH11 than the breast itself in normal condition [19]. Inhibition of CDH11 in breast cancer cell lines reduced migration and invasion ability [20], suggesting that it may play an important role in TNBC metastasis. 

In this present study, we hypothesized and validated our hypothesis that CDH11 regulates the canonical WNT signalling pathway and the CSCs-like phenotypes. The inhibition of the CSCs-like phenotypes of TNBC cells through CDH11 repression represents a novel therapeutic approach in TNBC treatment. Thus, we highlight a novel aspect of TNBC biology and provide a molecular basis for further exploration of CDH11-mediated anti-TNBC targeted therapy. 

## 2. Materials and Methods

### 2.1. Public Dataset Analyses

The TCGA Breast Cancer (BRCA) cohort datasets were downloaded from UCSC Xena Browser platform. This cohort contains several datasets from 1247 samples of breast cancer patients. We also made use of the TNBC cohort data of the Molecular Taxonomy of *Breast Cancer* International Consortium (METABRIC) cohort dataset (*n* = 1904) downloaded from the European Genome-Phenome archive (EGAS00000000098). The METABRIC study classifies breast tumors into subcategories, based on genetic fingerprints and molecular signatures which are intended to help predict therapeutic response and determine the optimal course of treatment. The gene expression RNAseq-IlluminaHiSeq and Phenotypes datasets were downloaded and used for further analysis. The PAM50 mRNA nature2012 clinical parameter was used for classifying breast cancer patients into luminal A, luminal B, Her2-enriched and basal-like (BL) subgroups. The status of ER, PR and Her2 were used to determine the triple negative breast cancer subgroup. To establish correlation between CDH11 and prognosis of breast cancer patient for each subgroup, we performed Kaplan Meier (KM) overall survival analysis using the “R2: Genomics Analysis and Visualization Platform”. For the low/high expression group dichotomization, we did not use the traditional median or mean cutoff values, rather we employed a bioinformatics approach using the automated ‘Kaplan scan’ cutoff function of the R2 genomic interface platform. The ‘Kaplan scan’ generates a KM plot based on the most optimal mRNA cut-off expression level to discriminate between a good (low expression) and bad (high expression) prognosis cohort. This was followed by the Bonferroni test for statistical significance (*p*-value) of the KM plot.

### 2.2. Immunohistochemistry

Triple negative breast cancer tissue samples (*n* = 38) were obtained from the Shuang Ho Hospital (SHH) breast cancer cohort. Ethical approval for the study was obtained from Joint Institutional Review Board (JIRB) of the Taipei Medical University (approval number: N201603028). Tissue sections (4 µm) were deparaffinized and rehydrated in gradually decreased concentration of methanol (100%, 95%, and 70%). Antigen retrieval was carried out by boiling slides in pressure cooker containing TrilogyTM buffer (Sigma-920P-06, Cell Marque, Sigma-Aldrich, Inc. St. Louis, MO, USA) for 5 min, and followed by incubation in hydrogen peroxide blocking solution (TA-125-H2O2Q, Thermo Fisher Scientific, Waltham, MA, USA) for 10 min. Nonspecific binding was blocked with Ultra V Block (TA-125-PBQ, Thermo Fisher Scientific, Waltham, MA, USA) for 10 min. The slides were incubated in primary antibodies against cadherin 11 (polyclonal antibody, 71-7600, Thermo Fisher Scientific, Waltham, MA, USA) and β-catenin (H-102: sc-7199, Santa Cruz Biotechnology, Santa Cruz, CA, USA) with working dilution 1:100 and 1:50, respectively for overnight at 4 °C. Later, tissue slides were incubated in Primary Antibody Amplifier Quanto (TL-125-QPB, Thermo Fisher Scientific, Waltham, MA, USA) for 10 min, in Horseradish peroxidase (HRP) Polymer Quanto (TL-125-QPH, Thermo Fisher Scientific, Waltham, MA, USA) for 10 min and then in DAB Quanto Chromogen (TA-004-QHCX, Thermo Fisher Scientific, Waltham, MA, USA) diluted 3:100 in DAB Quanto Substrate (TA-125-QHSX, Thermo Fisher Scientific, Waltham, MA, USA) for 3 min. Slides were counterstained with hematoxylin. The immunoreactive score system (IRS) was used to measure the expression level of protein of interest as previously described [21]. For final analyses, negative-mild staining was categorized as “Low” while moderate-strong positive staining was categorized as “High”. 

### 2.3. Cell Culture

TNBC cell lines, MDA-MB-231 and Hs578T were purchased from American Type Culture Collection (ATCC, Manassas, VA, USA). MDA-MB-231, derived from the pleural effusion and metastatic site of a female patient with breast adenocarcinoma, constitutively express WNT7B, EGF and TGFα, and forms poorly differentiated adenocarcinoma (grade III) in experimental mice models. Hs578T, however, is from a female patient with primary breast carcinoma and is non-tumorigenic in immunosuppressed mice. The selection of the 2 cell lines provided a basis for phenotypic and functional comparison between two variants of TNBC cells. The cell lines used in this study were periodically tested and confirmed to be free from mycoplasma and/or cross-contamination with cells derived for a different origin during laboratory manipulation or processing. Both cell lines were cultured in DMEM) (DML10-1000ML, Caisson Labs, Smithfield, UT, USA) supplemented with 10% heat-inactivated fetal bovine serum (FBS), 100 IU/mL Penicillin and 100 µg/mL Streptomycin, in 5% CO_2_ humidified atmosphere at 37 °C. For maintenance, both cell lines were sub-cultured every 48–72 h. 

### 2.4. CDH11 Knockdown

For CDH11 loss-of-function studies, we used On-Target plus Human CDH11 siRNA-Smart pool (L-013493-00-0005, Dharmacon, Lafayette, CO, US) that contains 4 siRNA mix. CDH11 #1: 5’-GUGAGAACAUCAUUACUUA-3’. CDH11 #2: 5’-GGACAUGGGUGGACACAUAUG-3’. CDH11 #3: 5’-GGAAAUAGCGCCAAGUUAG-3. CDH11 #4: 5’-CCUUAUGACUCCAUUCAAA-3’. A day before transfection, MDA-MB-231 cell line was seeded into 6-well cell plate at a density of 3 × 10^5^ cells/well in DMEM containing 10% FBS without antibiotics. The Hs578t cell line was seeded in the same condition except the density (2 × 10^5^ cells/well). After 24 hours incubation, 30–40% confluent cells were transfected with siRNA using Lipofectamine 3000 (L3000008, Thermo Fisher Scientific, Waltham, MA, USA) in serum-free DMEM. 80 nM of siRNA was used as final concentration. Six hours after initial transfection, the medium was replaced with fresh DMEM containing 10% FBS. Next day, cells were re-transfected with same siRNA, following same transfection steps. The siRNA-transfected cells were incubated for 48 h since the first transfection and used for western blot, and functional assays.

### 2.5. Western Blotting

Triple negative breast cancer cells were harvested by scraping method then lyzed with RIPA lysis buffer that supplemented with protease inhibitor (1X, Cat# 78430, Halt Protease Inhibitor Single-Use Cocktail, Thermo Fisher Scientific, Waltham, MA, USA) and phosphatase inhibitor (0.5X, Pierce^TM^ Phosphatase Inhibitor Mini Tablets, Thermo Fisher Scientific, Waltham, MA, USA). After homogenization with pellet pestle, the cells were incubated on ice for 30 min. The re-suspended cells were centrifuged at 12.000 rpm for 10 min; then, supernatants were collected as total protein samples. Pierce^TM^ BCA Protein Assay Kit (Thermo Fisher Scientific, Waltham, MA, USA) was used to determine final protein concentration. After boiling with sample buffer for 10 min and centrifuged for 10 min, the total protein samples were loaded onto and ran by sodium dodecyl sulfate-polyacrylamide gel electrophoresis (SDS-PAGE), then blots transferred onto polyvinylidene fluoride (PVDF) membranes, followed by blocking with 5% skimmed milk or 5% BSA in TBS-Tween-20 (TBST) for 1 h. The membranes were then incubated in primary antibodies against CDH11 (polyclonal antibody, 1:1000, Cat# 71-7600, Thermo Fisher Scientific, Waltham, MA, USA), β-catenin (monoclonal antibody, 1:1000, Cat# 9582P, Cell Signalling Technology, Danvers, MA, USA), c-Myc (monoclonal antibody, 1:500, Cat# sc-40, Santa Cruz Biothecnology, Santa Cruz, CA, USA), Met (monoclonal antibody, 1:1000, Cat# 8198, Cell Signalling Technology, Danvers, MA, USA), SOX2 (monoclonal antibody, 1:1000, Cat# 3579P, Cell Signalling Technology, Danvers, MA, USA), CD44 (monoclonal antibody, 1:1000, Cat# 3570, Cell Signalling Technology, Danvers, MA, USA), KLF4 (polyclonal antibody, 1:1000, Cat# 4038P, Cell Signalling Technology, Danvers, MA, USA), c-Jun (monoclonal antibody, 1:1000, Cat# 9165, Cell Signalling Technology, Danvers, MA, USA), MMP7 (monoclonal antibody, 1:1000, Cat# 3801, Cell Signalling Technology, Danvers, MA, USA),and β-actin (monoclonal antibody, 1:10.000, Cat# ab6276, Abcam Biotechnology Company, Cambridge, MA, UK) at 4 °C overnight. After washing with TBST, membranes were probed with an HRP-labeled anti-rabbit or anti-mouse IgG secondary antibody as appropriate. Protein bands were visualized by UVP Biospectrum Imaging System (Vision Works LS 6.8, Level Biotechnology Inc. New Taipei City, Taiwan) using ECL reagents (Thermo Fisher Scientific, Waltham, MA, USA).

### 2.6. Transwell Matrigel Invasion and Migration Assay

We used the 24-well plate transwell system to evaluate migration ability. The upper chambers of the transwell were pre-coated with solubilized Matrigel (BD Bioscience). After the matrigel polymerized, MDA-MB-231 and Hs578t cell lines (2 x 10^4^ cells/well) were seeded into the upper chamber containing 200 μL serum-free DMEM, while 500 μL DMEM with 10% FBS in the lower chamber served as chemoattractant. The medium was discarded after 24 h incubation and cells on the membrane were fixed using formaldehyde (10%) for 20 min at room temperature before crystal violet staining for 20 min. The cells on the upper side of the membrane were carefully removed with a cotton swab. The migrated cells were observed under microscope and the total number of cells on the lower surface counted. Migration assay was performed following same steps as the invasion assay, but without Matrigel on the upper chamber.

### 2.7. Immunofluorescence Assay

The wild-type and siCDH11 MDA-MB-231 and Hs578t cells were seeded in 96-well plates and incubated for 24 h, washed with 200 μL ice-cold PBS, then fixed using 200 μL pre-chilled 4% formaldehyde for 15 min at room temperature before being blocked with 200 μL blocking buffer (1x PBS/ 5% BSA/ 0.3% Triton^TM^ X-100) for 1 h at room temperature. After washing with PBS, the cells were incubated with 100 μL primary antibodies against CDH11 (monoclonal antibody, Cat#, 5B2H5, 1:50, Thermo Fisher Scientific, Waltham, MA, USA), or β-catenin (monoclonal antibody, Cat# 9582P, 1:50, Cell Signalling Technology, Danvers, MA, USA) at 4 °C overnight in a dark room. Next day, cells were washed with 0.1% PBS, incubated with 100 µL Alexa-Flour-conjugated secondary antibodies against rabbit and mouse for 1 h at room temperature in the dark. In this experiment, the primary and secondary antibodies were diluted in antibody dilution buffer (1x PBS/ 1% BSA/ 0.3% Triton^TM^ X-100). Next, cells were washed with PBS and were counterstained with DAPI (0.5 μg/mL, Vector Laboratories, Burlingame, CA, USA) and evaluated using fluorescence microscope. 

### 2.8. Mammosphere Formation Assay

The wild-type and siCDH11 MDA-MB-231 and Hs578t cells were harvested and seeded in 24-well plates (Cat# CLS3473, Corning Costar Ultra-Low Attachment Multiple Well Plates, Sigma-Aldrich, St. Louis, MO, USA) that contained 500 μL serum-free DMEM medium, supplemented with B27 (Invitrogen, Carlsbad, CA, USA), and 10 ng/mL epidermal growth factor (BD Biosciences, Palo Alto, CA, USA). The cells were seeded in 3 different density (1000 cells/cm^2^, 2000 cells/cm^2^, and 4000 cells/cm^2^) for each group. After 5 days incubation, formed mammospheres were observed and evaluated under microscope. 

### 2.9. In Vivo Study

The MDA-MB-231 and Hs578t cells were treated with siRNA specific for CDH11 for 48h. NOD/SCID mice were randomized into wild type (*n* = 5) and siCDH11 (*n* = 5) group for each cell line. The mice were inoculated subcutaneously with 2 × 10^6^ wild type (WT) or siCDH11 MDA-MB-231 or Hs578t cells in their hind flank. Mice tumor sizes were measured on days 6, 9, 12, 15, 18, and 27 after TNBC cell inoculation using callipers and tumor volumes calculated with a standard formula: length × width^2^ × 0.5. The tumor-bearing mice were sacrificed, and the tumor mass were observed and measured on day 27 post-inoculation. In parallel in vivo studies following same steps except sacrificing on day 27, the mice were observed until day 45 post-inoculation to assess the effect of altered CDH11 expression on the survival rates. For maintenance of siRNA effect, mice in the siCDH11 group were injected intra-tumorally with 10 μmol/L of ‘siCDH11-atelocollagen’ complex on days 9, 18, 27, and 36 after TNBC cell inoculation. The siCDH-atelocollagen complex, usually prepared the preceding evening and stored at 4 °C before use, consisted of equal volumes of well mixed siCDH11 and atelocollagen (Koken Co. Ltd., Tokyo, Japan). All tumor xenograft animal studies were approved by the TMU-SHH Joint Institutional Review Board and performed in accordance with protocol approved by the TMU Institutional Animal Care and Use Committee (approval number: LAC-2015-0383).

### 2.10. Statistical Analysis

Kaplan Meier overall survival analysis were performed using “R2: Genomics Analysis and Visualization Platform” and GraphPad Prism for Windows version 7.00 software (GraphPad Software Inc., San Diego, CA, USA). Statistical significance was assumed when *p* ≤ 0.05. For correlation studies between CDH11 and several components of the WNT signalling pathway SigmaPlot for Windows version 10.0 software (Systat Software Inc., San Jose, CA, USA) was used. All cell-based assays were performed at least three times in triplicates and results expressed as Mean ± SD. Quantitative and qualitative analyses of western blot, mammosphere formation, migration and invasion assays data were performed using ImageJ version 1.51j8 (Wayne Rasband National Institutes of Health, Bethesda, MD, USA) and the graph results were created using SigmaPlot.

## 3. Results

### 3.1. High CDH11 Expression has Significant Positive Correlation with Poor Overall Survival in Patients with Basal-Like and Triple Negative Breast Cancer

To determine existing correlation or association between CDH11 expression and prognosis of breast cancer patients, we assessed TCGA Breast Cancer (BRCA) cohort datasets from UCSC Xena Browser (https://xenabrowser.net/datapages/). We divided the breast cancer patients into luminal A, luminal B, Her2-enriched, BL and TNBC subgroup. We observed that high CDH11 expression correlated with significantly better survival rates in the luminal A (*p* = 0.0381) and Her-2 enriched (*p* = 0.0319) subgroups, while no statistically significant correlation was observed between CDH11 expression and luminal B breast cancer patient survival (*p* = 0.2831) (Figure 1A–C). However, in contrast, high CDH11 expression in the BL (*p* = 0.0049) and TNBC (*p* = 0.045) subgroups significantly correlated with worse overall survival (Figure 1D,E). Our data is consistent with the aggressive phenotype of TNBC cells and their poor prognosis despite initial good response to therapy and corroborate significant overlap between the TNBC and BL breast cancer molecular subtypes [22,23]. These results indicate the prognostic nature of CDH11 expression profile and suggest its role as a potential candidate for anti-TNBC targeted therapy.

### 3.2. CDH11 Positively Modulates β-Catenin Expression and Is Associated with Activation of the Canonical WNT Signalling Pathway

β-catenin is a key component of the canonical WNT signalling pathway and has been shown to directly bind to the cytoplasmic domain of CDH11 (13). We hypothesized that CDH11 plays an important role in enhancing the stemness, migration, and invasion potential of TNBC cells through activation of the canonical WNT signalling pathway. Using the METABRIC-Breast Cancer (IlluminaHiSeq) cohort dataset (*n* = 1904), we investigated the correlation between CDH11 and several components of the WNT signalling pathway in the TNBC subgroup. We found that CDH11 expression positively correlated with β-catenin, WNT2, and TCF2 expression with pearson’s r = 0.29 (*p* = 8.3 × 10^−30^), 0.60 (*p* = 7.8 × 10^−183^) and 0.48 (*p* = 1.3 × 10^−101^), respectively (Figure 2A–C). Corroborating the results above, our IHC data demonstrated that tissues from TNBC patients concurrently overexpressed CDH11 and β-catenin proteins, compared to the weak/non-expression in non-tumor tissues (Figure 2D–F). In fact, correlative analysis of our local TNBC cohort (*n* = 38) indicate a strong positive correlation between CDH11 and β-catenin (r = 0.62; *p* < 0.0001; Figure 2F, *lower panel*). To reconfirm these findings and establish the functional significance of CDH11 expression and/or activity in TNBC cells, we knocked-down CDH11 in MDA-MB-231 and Hs578t TNBC cell lines and examined its effect on WNT signalling using western blot and immunofluorescence staining assays. Reduced CDH11 expression in the TNBC cells altered their morphology from spindle-like to polygonal, reduced cell viability, and increased cell doubling time (Figure 3A,B). Additionally, our western blot data showed that MDA-MB-231 and Hs578t cells transfected with siCDH11 exhibited significantly decreased expression levels of CDH11, β-catenin, Met, c-Myc, c-Jun, and MMP7 (Figure 3C,D). Furthermore, we observed that siCDH11 in MDA-MD-231 and Hs578t cells reduced β-catenin nuclear localization and expression (Figure 4). These data corroborate the modulatory role of CDH11 on the WNT/β-catenin signalling pathway and suggest that CDH11 plays an important role in the maintenance of β-catenin protein in the MDA-MB-231 and Hs578t TNBC cell lines.

### 3.3. CDH11 Inhibition Suppresses the Stem Cell-Like Phenotype of TNBC Cells

CSCs constitute a subset of cancer cells with self-renewal ability and enhanced resistance to some chemotherapy agents [6]. We investigated whether CDH11 plays an important role in TNBC stemness. Tumorsphere formation assays using wild-type (WT) and siCDH11 MDA-MB-231 and Hs578t cells show that the siCDH11 cells displayed lesser tumorsphere formation efficiency compared to the WT cells (Figure 5A–E). Measurement of tumorsphere sizes showed that all the siCDH11 tumorspheres were significantly smaller than those formed by their WT counterparts of both cell lines except in the lowest seeding cells of Hs578t (Figure 5B,C). The siCDH11 cells produced fewer tumorspheres than WT cells at all seeding density, except 4000 cells/cm^2^ (Figure 5D,E). To further confirm the important role of CDH11 in TNBC stemness, we assessed several CSCs markers. We observed significantly downregulated Sox2, KLF4, CD44, and c-Myc protein expression levels in the siCDH11 group compared to the WT group (Figure 5F and Appendix A). These data demonstrate that tumorsphere formation efficiency is suppressed in the CDH11-deficient MDA-MB-231 and Hs578t TNBC cell lines and suggest a role for CDH11 expression in the modulation of the stem cell-like phenotype of TNBC cells.

### 3.4. CDH11 Inhibition Markedly Attenuate the Migration And invasion of TNBC Cells

Since the CSCs-like phenotype is often associated with enhanced oncogenicity and metastatic traits [24], we further examined whether and how alteration in CDH11 expression affects the characteristic aggressiveness of TNBC cells, using WT and siCDH11 MDA-MB-231 and Hs578t cells. We demonstrated that the siCDH11 cells exhibited significantly reduced migration and invasion ability; fewer migrated siCDH11 MDA-MB-231 and Hs579t cells were observed than for the WT cells (Figure 5G). Similarly, siCDH11 resulted in significantly less invaded cells, compared to the WT cells (Figure 5G and Appendix A). Since, cell migration and invasion are objective measures of the aggressiveness of cancerous cells; these results further corroborate the vital role of CDH11 in the enhanced metastatic phenotype and aggressiveness of TNBC cells.

### 3.5. Silencing CDH11 Significantly Attenuates Tumorigenicity and Tumor Growth of TNBC Cells, In Vivo

Having previously demonstrated that silencing CDH11 suppresses the stem cell-like phenotype of TNBC cells in vitro, to determine the probable inhibitory effect of silencing CDH11 on tumor formation and growth in vivo, we generated tumor xenograft models derived from NOD/SCID mice (*n* = 5/group) inoculated with 2 × 10^6^ WT or siCDH11 MDA-MB-231 or Hs578t cells subcutaneously in the hind-flank and observed for 27 days (Figure 6A). We demonstrated that the mice inoculated with siCDH11 MDA-MB-231 or Hs578T developed very significantly smaller tumors, compared to the WT group (MDA-MB-231: ~6.58-fold smaller, *p* = 0.008; Hs578t: ~5.75-fold smaller, *p* = 0.003 on day 27) (Figure 6B,C). siRNA effect in the siCDH11 group was maintained by intra-tumoral injection with 10 μmol/L of ‘siCDH11-atelocollagen’ complex (Appendix A). Of translational relevance, in parallel mice assays, survival analyses show that over 45 days, mice bearing siCDH11 MDA-MB-231 or Hs578T cells-derived tumors enjoyed 20–50% (*p* = 0.027) or 16–50% (*p* = 0.011) survival advantage, respectively, compared to their WT-inoculated litter-mate counterparts (Figure 6D,E). Taken together, these findings suggest that CDH11 play an important role in tumorigenicity and tumor growth of TNBC cells in vivo. As depicted in the schematic abstract (Figure 6F), these data showing siCDH11-associated impaired tumorigenesis and increased survival substantiate CDH11-targeting as a potential therapeutic strategy for patients with TNBC.

## 4. Discussion

Cadherin 11 is a transmembrane protein with documented conflicting roles in several types of cancers [25,26,27,28]. Recently, it was suggested that CDH11 acts as a tumor suppressor gene as demonstrated by its methylated and silenced status in malignant tissues such as in the nasopharyngeal, esophageal, gastric, hepatocellular carcinoma, colon, and breast carcinoma [26]. Similarly, in vitro and in vivo studies of murine retinoblastoma, cdh11 acted as a tumor suppressor gene through promotion of tumor cell death [29]. Conversely, in ovarian cancer patients, CDH11 exhibited no evidence of tumor-suppressive or oncogenic function, nor bore any prognostic / predictive relevance [27]. However, cumulative evidence supports the role of cdh11 as an oncogene, including that reported in the present study. Higher CDH11 mRNA expression level has been shown in malignant breast tumor than in non-malignant breast tumor or normal breast tissues [14]. Another study showed that increased expression of CDH11 is characteristic of enhanced invasiveness in breast cancer cell lines, in vitro [30]. In the present study, we demonstrated that high CDH11 expression positively correlates with poor prognosis in the more aggressive breast cancers subtypes, namely BL and TNBC subtypes (Figure 1). Thus, we posit that the high expression of CDH11 in patients with BL breast cancer and TNBC indicates that CDH11 plays an important role in these breast cancer molecular subtypes as an oncogene. This is consistent with the findings from prostate and renal cancers, wherein tissue samples from the bone metastatic site exhibited higher expression levels of the CDH11 protein than observed in primary tumor or adjacent normal tissue [15,16].

Based on breast cancer big data analyses, we showed that CDH11 expression positively correlates with expression of WNT signalling components such as β-catenin, WNT2, and TCF2, and that tissues from TNBC patients show concurrent high expression of CDH11 and β-catenin proteins; thus, we demonstrate that CDH11 positively modulates β-catenin expression and activates the canonical WNT signalling pathway (Figure 2). These findings are consistent with contemporary knowledge that β-catenin is a core mediator of the canonical WNT signalling pathway. In the presence of WNT molecule, β-catenin translocates from the cytoplasm to the nucleus, induces downstream target genes, subsequently activating the canonical WNT pathway [9]; however, with absent or deactivated WNT molecule, cytoplasmic β-catenin binds to the APC-Axin-CK1-GSK3β complex, otherwise called destruction complex, leading to β-catenin phosphorylation [11]. The phosphorylated β-catenin is degraded in the proteasome, thus, reducing β-catenin bioavailability [11]. Cadherin acts as a reservoir for calcium-dependent adhesion-competent β-catenin when WNT signalling is inactivated [13]. 

Validating our hypothesis that CDH11 may promote β-catenin nuclear translocation and WNT activation, we demonstrated that CDH11 modulates β-catenin expression levels and that CDH11/β-catenin signalling axis plays a regulatory role for the canonical WNT signalling pathway in TNBC (Figure 3), as concomitant decreased CDH11 and β-catenin in the TNBC cell was observed, with siCDH11 cells exhibiting reduced CDH11/β-catenin nuclear co-localization compared to the control WT cells (Figure 4). This is consistent with studies suggesting that the transcriptional suppression of β-catenin was associated with inhibition of the WNT signalling activity in MCF-4 and A549 cell lines [31]. Based on our immunofluorescence and IHC staining, CDH11 localizes in the membrane and cytoplasm. In rheumatoid arthritis, matrix metalloproteinases (MMPs)- and γ-secretase- induced cleavage induced CDH11 activity, through its release from the cytoplasmic domain and subsequent nuclear translocation [32,33]. This in-part explains our observed nuclear co-localization of CDH11 and β-catenin and subsequent WNT signalling activation.

Aside possessing the ability to self-renew and differentiate into heterogeneous tumor cells, CSCs are implicated in the drug-resistance, relapse, distant metastasis and poor prognosis in TNBC patients [34,35]. We demonstrated that CDH11 knockdown significantly reduces the size of formed mammospheres, with concomitant downregulation of CSCs markers, Sox2, KLF4, CD44, and c-Myc (Figure 5), which is consistent with previous findings showing that inhibition of the canonical WNT signalling decreased Sox2 [36], CD44 [24], and c-Myc, as well as suppressed tumorsphere formation and inhibited tumor formation in xenograft tumor mice models [34,37]. Taken together, our results highlight a modulatory role for the CDH11/β-catenin signalling axis in TNBC CSCs-like. 

Having established the functional relevance of the CDH11/β-catenin signalling axis in TNBC CSCs-like phenotype, we further confirmed that this axis modulates TNBC metastatic phenotype as demonstrated by markedly suppressed migration and invasion ability of CDH-deficient TNBC cells (Figure 5). This finding is consistent with the previously documented role of CDH11 in the promotion of migration and invasion in colorectal and prostate cancer [38,39]. Finally, substantiating findings from our in vitro assays, our in vivo studies highlight the putative potential of CDH11-targeting to suppress tumorigenesis, and inhibit tumor growth in TNBC xenograft models, while concurrently improving survival time (Figure 6 and Appendix A). 

## 5. Conclusions

In summary, our findings highlight the critical roles CDH11 plays in TNBC. We demonstrate for the first time, to the best of our knowledge, that by targeting β-catenin, CDH11 regulates the canonical WNT signalling pathway, inhibits the CSCs-like and metastatic phenotypes of TNBC cells, and represents a novel therapeutic approach in TNBC treatment. We herein provide a basis for further exploration of CDH11 as a putative candidate for targeted therapy in triple negative breast cancer.

## Figures and Tables

**Figure 1 jcm-08-00148-f001:**
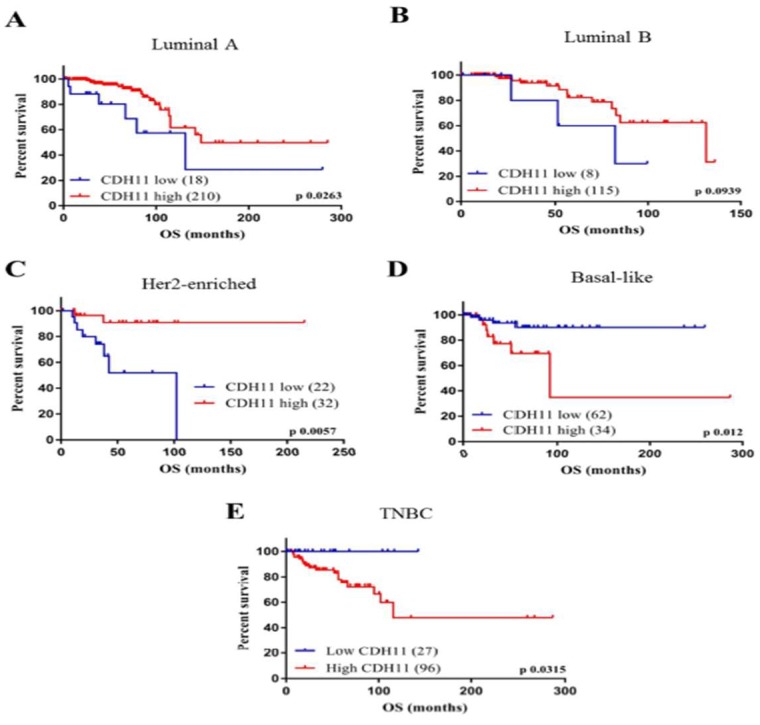
Patients with TNBC and basal-like breast cancer exhibiting high expression of CDH11 are characterized by worse prognosis. Kaplan-Meier plots based on analyses of breast cancer cohort (*n* = 1247) using the TCGA Breast Cancer (BRCA) cohort datasets show the effect of CDH11 expression on the overall survival rates in patients regardless of molecular subtype, (**A**) Luminal A, (**B**) Luminal B, (**C**) Her2-enriched, (**D**) Basal-like, or (E) Triple negative breast cancer.

**Figure 2 jcm-08-00148-f002:**
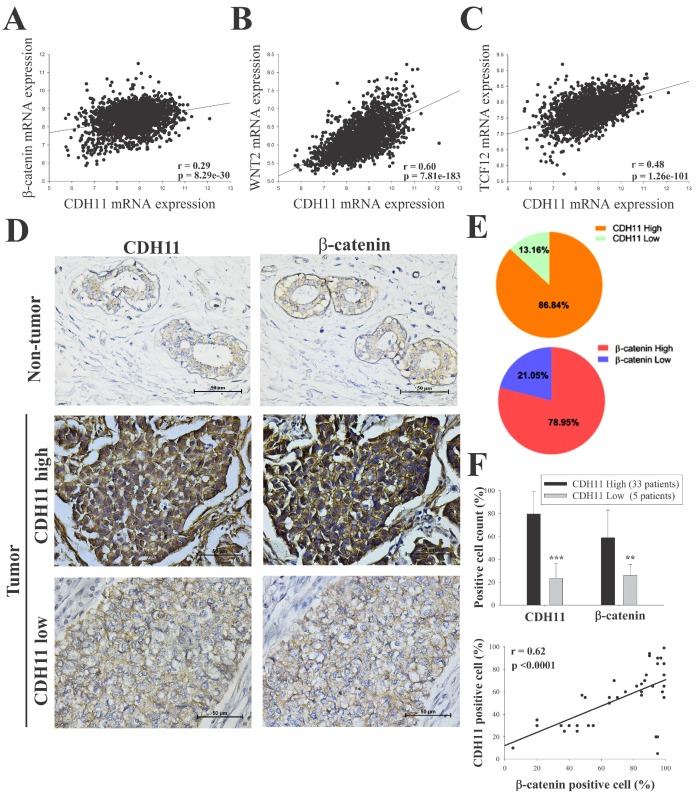
CDH11 positively modulates β-catenin expression and activates the canonical WNT signalling pathway. Statistical analyses of the TNBC sub-group of the METABRIC Breast Cancer cohort (*n* = 1904) show positive correlation between expression of CDH11 and (**A**) β-catenin, (**B**) WNT2, and (**C**) TCF12 mRNA. (**D** and **E**) Differential expression of CDH11 and β-catenin in non-tumor and TNBC tissue samples demonstrated by representative IHC images and percentages of patients. Scale bar: 50 μm. (**F**) Histogram showing positive correlation between CDH11 and β-catenin protein expression level in our TNBC cohort (*n* = 38). * *p* < 0.05, ** *p* < 0.01, *** *p* < 0.001

**Figure 3 jcm-08-00148-f003:**
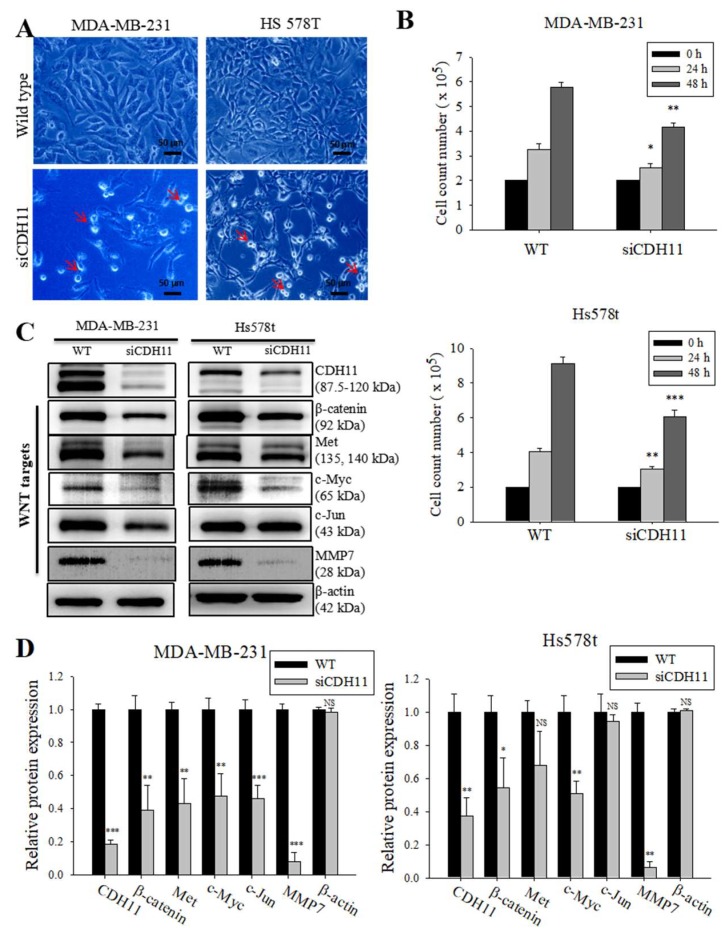
Loss of CDH11 function using siRNA changes cell morphology and inhibits activation of the canonical WNT signalling pathway in TNBC cells. (**A**) The cell morphology of MDA-MB-231 and Hs578t changed from spindle-like to polygonal after loss of CDH11 expression. (**B**) Graphical representation of the effect of siCDH11 on the doubling time of MDA-MB-231 and Hs578t cells as determined by trypan-blue exclusion. (**C**) Western blot results show CDH11 knockdown decreased β-catenin, Met, c-Myc, c-Jun, and MMP7 protein expression level in MDA-MB-231 and Hs578t cell lines. (**D**) Graphical representation of **C**. Results expressed as mean ± SD of assays done 3 times in triplicate. * *p* < 0.05, ** *p* < 0.01, *** *p* < 0.001; NS, not significant; Red arrow, floating non-viable cells.

**Figure 4 jcm-08-00148-f004:**
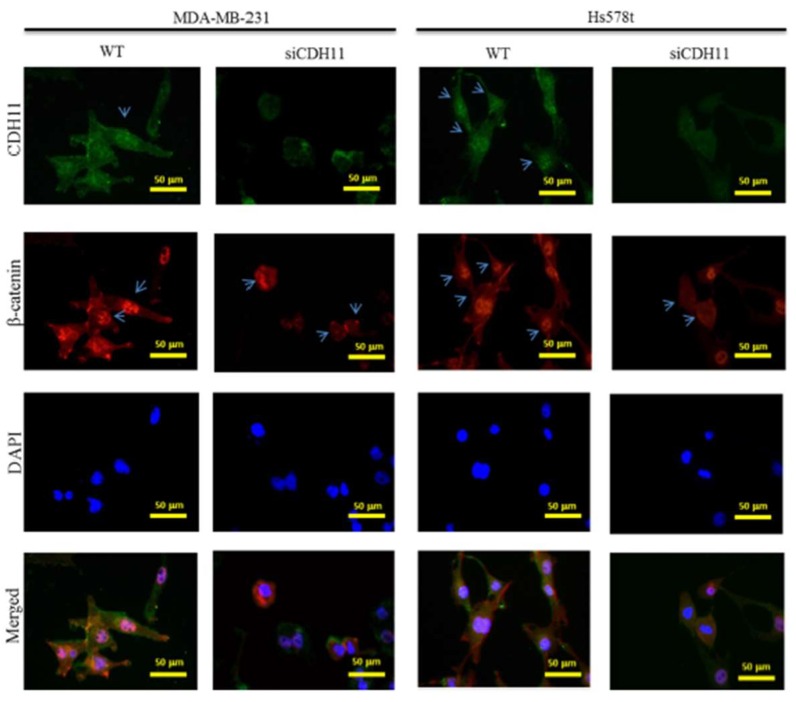
Knockdown of CDH11 using siRNA reduces β-catenin nuclear co-localization in TNBC cells. Immunofluorescence staining of MDA-MB-231 and Hs578t cells with CDH11 (green), β-catenin (red), and DAPI (blue). The results show that silencing CDH11 concurrently reduced CDH11 and β-catenin protein expression level and their nuclear localization. The blue arrows indicate presence and absent of CDH11 and β-catenin in the nuclear region. Images are representative of four separate assays in triplicate. Scale bar: 50 μm.

**Figure 5 jcm-08-00148-f005:**
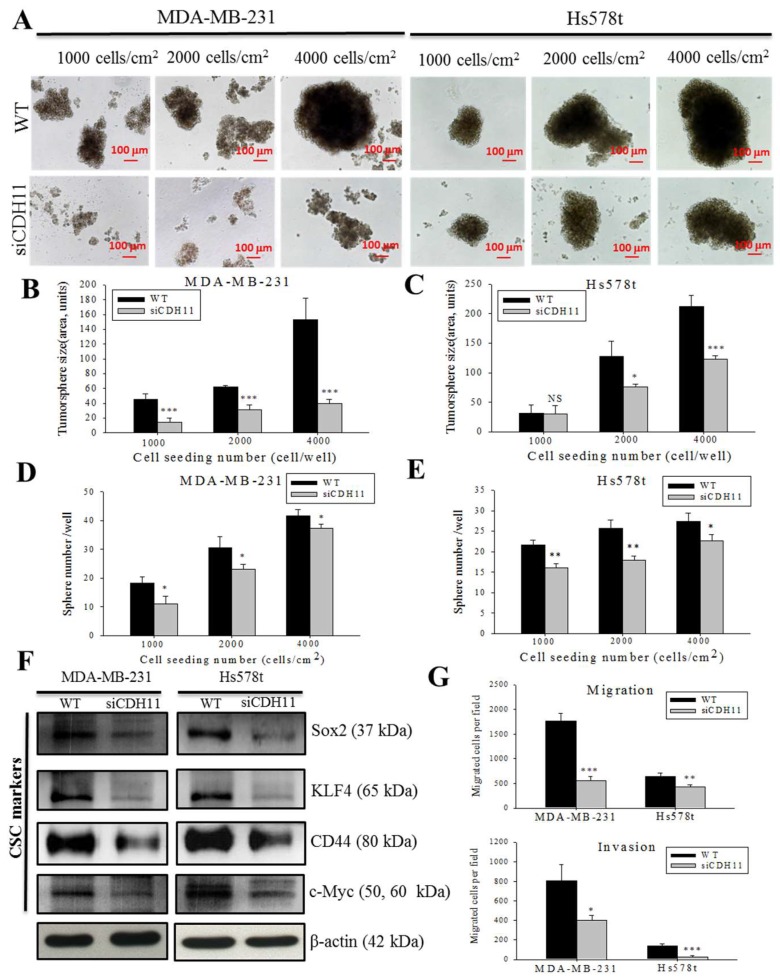
CDH11 inhibition suppresses the stem cell-like phenotype of TNBC cells. (**A**) Images showing tumorsphere size in wild-type cells and their siCDH11 counterparts of varying cell density. Histograms comparing the tumorsphere sizes of wild-type or si-CDH11 (**B**) MDA-MB-231 and (**C**) Hs578t. Histograms comparing the number of tumorspheres formed by wild-type or si-CDH11 (**D**) MDA-MB-231 and (**E**) Hs578t cells, based on ImageJ quantification. Only tumor spheres ≥40 μm in size were counted. (**F**) The effect of siCDH11 on the expression Sox2, KLF4, CD44, and c-Myc proteins in MDA-MB-231 and Hs578t cells. (**G**) Graphical representation of the migration (*upper panel)* and invasion *(lower panel)* of MDA-MB-231 and Hs578t cells. Results represent mean ± SD of 3 independent assays in triplicate. * *p* < 0.05, ** *p* < 0.01, *** *p* < 0.001; WT, wild type; NS, not significant; Scale bar: 100 μm.

**Figure 6 jcm-08-00148-f006:**
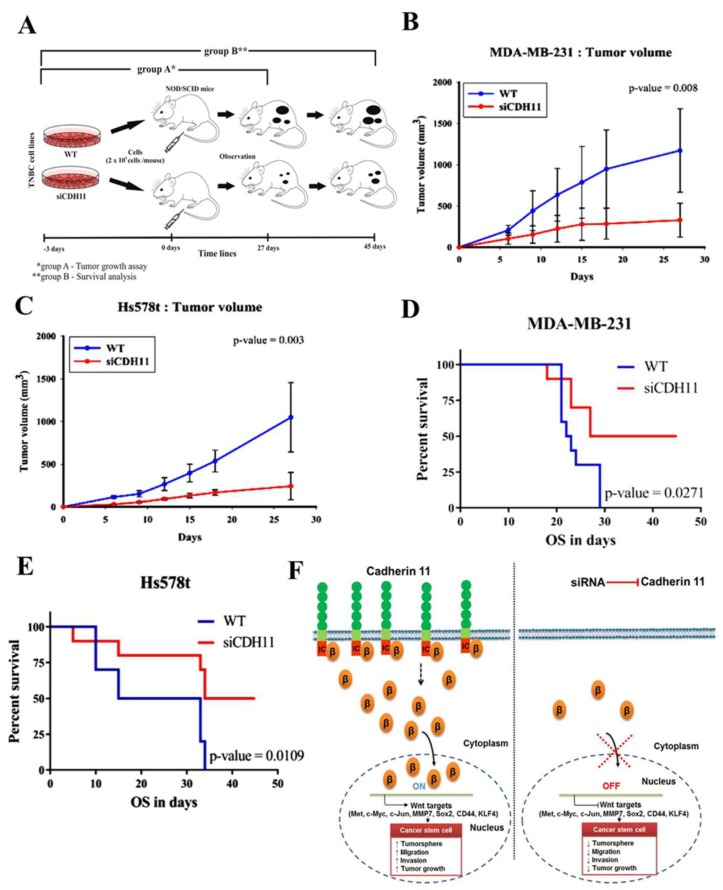
Silencing CDH11 significantly attenuates tumorigenicity of TNBC cells, in vivo. (**A**) Schema showing xenografts derived from NOD/SCID mice injected with 2 × 10^6^ WT or siCDH11 MDA-MB-231 or Hs578t cells subcutaneously in the hind-flank and observed for 27 days (group A, *n* = 5/group) or 45 days (group B, *n* = 5/group). Graph showing difference in tumor volume over time for the WT or siCDH11 (**B**) MDA-MB-231 or (**C**) Hs578T tumor-bearing mice. Intergroup *p*-values were determined by 2-way ANOVA. Kaplan-Meier survival curves for mice bearing WT or siCDH11 (**D**) MDA-MB-231 or (**E**) Hs578T cell-derived tumors. WT, wild type. (**F**) Schematic abstract showing that CDH11 inhibition suppresses the TNBC CSCs-like and metastatic phenotypes through WNT signalling regulation. In the presence of WNT signal, cytoplasmic β-catenin is released from CDH11, accumulates in the perinuclear region, then translocates to the nucleus where it activates downstream CSCs-regulating target genes, including Met, c-Myc and Sox2 by binding to TCF/LEF. Once activated, these genes enhance self-renewal, migration, and invasion abilities of TNBC cells. In the absence of CDH11, membranar, cytoplasmic and nuclear β-catenin pool is diminished, WNT signalling is obtused, and the canonical WNT signalling pathway downstream target genes remains inactivated.

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
