# Peer review of "Cadherin 11 Inhibition Downregulates β-catenin, Deactivates the Canonical WNT Signalling Pathway and Suppresses the Cancer Stem Cell-Like Phenotype of Triple Negative Breast Cancer"

_jcm, 2019, doi:10.3390/jcm8020148_

Reviewer 1 Report

The authors have tested the correlation between Cadherin 11 and Beta-catenin Wnt signaling pathway in triple negative breast cancers. The authors need to reword some of their conclusions and interpret their data conservatively. I have listed my suggestions below:

1.     The paragraph in introduction from line 58 needs to be rewritten. It is difficult to follow. I would suggest the authors keep the sentences short.

2.     In figure 2, there is a positive correlation between Wnt2 and Cdh11 but the correlation between Cdh11 and Beta-Catenin/TCF2 is very weak. This should be stated in the text. There is no correlation or weak correlation between Cdh11 and Beta-Catenin/TCF2.

3.     In line 289, authors should state why they used MDA-MB-231 and Hs578t cell lines. Simply stating that these are two breast cancer cell lines and briefly explaining their advantages relevant to this study is sufficient.

4.     In all the figures, text in the graphs and images needs to be bigger.     

5.     In IHC experiment in figure 2, TNBC patient samples should be compared to normal patient samples. Has this been done?

6.     Line 298 should state that Cdh11 plays an important role in the maintenance of beta-catenin protein on the surface in MDA-MB-231 and Hs578t cell lines.

7.     In figure 5F, only Klf4 and c-Myc are significantly decreased from the blots. Sox2 is only decreased in the MDA cell line. Cd44 levels do not look decreased. The authors need to correct their interpretation of the data.

8.     In lines 341-343, the authors use TNBC cells. This should be changed to the specific cell lines they  have used (MDA and hs578t) or they should state TNBC cell lines.

9.     In line 444, expression should be changed to levels.

Author Response

Answers to the comments

Point-by-point responses to reviewer’s comments:

We would like to thank the reviewer for the thorough reading of our manuscript as well as their valuable comments. We have followed their comments closely and feel that they have further strengthened the manuscript. Below are our point-by-point responses.

Q1: Reviewer #1:  The authors have tested the correlation between Cadherin 11 and Beta-catenin Wnt signaling pathway in triple negative breast cancers. The authors need to reword some of their conclusions and interpret their data conservatively. I have listed my suggestions below:

A1: We sincerely thank the reviewer for the time taken to review our work and the important suggestions made.

Q2: Reviewer #1: The paragraph in introduction from line 58 needs to be rewritten. It is difficult to follow. I would suggest the authors keep the sentences short.

A2: We thank the reviewer for this suggestion. We have made the change as requested by the reviewer. Please kindly see Introduction section, page 3, lines 58-59.

Wnt signalling pathway is a principal regulator of self-renewal, and helps maintain the undifferentiated state of normal mesenchymal stem cells and CSCs (7,8).

Q3: Reviewer #1: In figure 2, there is a positive correlation between Wnt2 and Cdh11 but the correlation between Cdh11 and Beta-Catenin/TCF2 is very weak. This should be stated in the text. There is no correlation or weak correlation between Cdh11 and Beta-Catenin/TCF2.

A3: We thank the reviewer for this very important observation. Please kindly see our revised Results section, page 11, lines 285 – 312.

CDH11 positively modulates β-catenin expression and activates the canonical WNT signalling pathway

β-catenin is a key component of the canonical WNT signalling pathway, and has been shown to directly bind to the cytoplasmic domain of CDH11 (13). We hypothesized that CDH11 plays an important role in enhancing the stemness, migration, and invasion potential of TNBC cells through activation of the canonical WNT signalling pathway. Using the METABRIC-Breast Cancer (IlluminaHiSeq) cohort dataset (n = 1904), we investigated the correlation between CDH11 and several components of the WNT signalling pathway in the TNBC subgroup. We found that CDH11 expression positively correlated with β-catenin, WNT2, and TCF2 expression with pearson’s r = 0.29 (p = 8.3e-30), 0.60 (p = 7.8e-183) and 0.48 (p = 1.3e-101), respectively (Figure 2A-C). Corroborating the results above, our IHC data demonstrated that tissues from TNBC patients concurrently overexpressed CDH11 and β-catenin proteins, compared to the weak/non-expression in non-tumor tissues (Figure 2D-F). In fact, correlative analysis of our local TNBC cohort (n = 38) indicate a strong positive correlation between CDH11 and b-catenin (r = 0.62; p<0.0001; Figure 2F, lower panel). To reconfirm these findings and establish the functional significance of CDH11 expression and/or activity in TNBC cells, we knocked-down CDH11 in MDA-MB-231 and Hs578t TNBC cell lines and examined its effect on WNT signalling using western blot and immunofluorescence staining assays. Reduced CDH11 expression in the TNBC cells altered their morphology from spindle-like to polygonal, reduced cell viability, and increased cell doubling time (Figure 3A). Additionally, our western blot data showed that MDA-MB-231 and Hs578t cells transfected with siCDH11 exhibited significantly decreased expression levels of CDH11, β-catenin, Met, c-Myc, c-Jun, and MMP7 (Figures 3B-D). Furthermore, we observed that siCDH11 in MDA-MD-231 and Hs578t cells reduced β-catenin nuclear localization and expression (Figure 4). These data corroborate the modulatory role of CDH11 on the Wnt/β-catenin signalling pathway and suggest that CDH11 plays an important role in the maintenance of β-catenin protein in the MDA-MB-231 and Hs578t TNBC cell lines.

Also kindly see our new Figure 2 and its legend, Page 11, Lines 315 – 321.  

Figure 2. CDH11 positively modulates β-catenin expression and activates the canonical WNT signalling pathway. Statistical analyses of the TNBC sub-group of the METABRIC Breast Cancer cohort (n = 1904) show positive correlation between expression of CDH11 and (A) β-catenin, (B) WNT2, and (C) TCF2. (D and E) Differential expression of CDH11 and β-catenin in non-tumor and TNBC tissue samples demonstrated by representative IHC images and percentages of patients. Scale bar: 50 μm.  (F) Histogram showing positive correlation between CDH11 and β-catenin expression level in our TNBC cohort (n = 38).

Q4: Reviewer #1: In line 289, authors should state why they used MDA-MB-231 and Hs578t cell lines. Simply stating that these are two breast cancer cell lines and briefly explaining their advantages relevant to this study is sufficient.

A4: We appreciate the reviewer’s comments. We have incorporated the characteristics of the selected TNBC cell lines in the revised manuscript. Please kindly see Materials & Methods section, Page 5, Lines 126 – 140.

Cell culture

TNBC cell lines, MDA-MB-231 and Hs578T were purchased from American Type Culture Collection (ATCC, Manassas, VA, USA). MDA-MB-231, derived from the pleural effusion and metastatic site of a female patient with breast adenocarcinoma, constitutively express WNT7B, EGF and TGFa, and forms poorly differentiated adenocarcinoma (grade III) in experimental mice models. Hs578T, however, is from a female patient with primary breast carcinoma and is non-tumorigenic in immunosuppressed mice. The selection of the 2 cell lines provided a basis for phenotypic and functional comparison between two variants of TNBC cells. The cell lines used in this study were periodically tested and confirmed to be free from mycoplasma and/or cross-contamination with cells derived for a different origin during laboratory manipulation or processing. Both cell lines were cultured in DMEM) (DML10-1000ML, Caisson Labs, USA) supplemented with 10% heat-inactivated fetal bovine serum (FBS), 100 IU/ml Penicillin and 100 µg/ml Streptomycin, in 5% CO2 humidified atmosphere at 37oC. For maintenance, both cell lines were sub-cultured every 48-72 h.

Also, kindly see Results section, Page 11, Lines 285 – 312.

CDH11 positively modulates β-catenin expression and activates the canonical WNT signalling pathway

β-catenin is a key component of the canonical WNT signalling pathway, and has been shown to directly bind to the cytoplasmic domain of CDH11 (13). We hypothesized that CDH11 plays an important role in enhancing the stemness, migration, and invasion potential of TNBC cells through activation of the canonical WNT signalling pathway. Using the METABRIC-Breast Cancer (IlluminaHiSeq) cohort dataset (n = 1904), we investigated the correlation between CDH11 and several components of the WNT signalling pathway in the TNBC subgroup. We found that CDH11 expression positively correlated with β-catenin, WNT2, and TCF2 expression with pearson’s r = 0.29 (p = 8.3e-30), 0.60 (p = 7.8e-183) and 0.48 (p = 1.3e-101), respectively (Figure 2A-C). Corroborating the results above, our IHC data demonstrated that tissues from TNBC patients concurrently overexpressed CDH11 and β-catenin proteins, compared to the weak/non-expression in non-tumor tissues (Figure 2D-F). In fact, correlative analysis of our local TNBC cohort (n = 38) indicate a strong positive correlation between CDH11 and b-catenin (r = 0.62; p<0.0001; Figure 2F, lower panel). To reconfirm these findings and establish the functional significance of CDH11 expression and/or activity in TNBC cells, we knocked-down CDH11 in MDA-MB-231 and Hs578t TNBC cell lines and examined its effect on WNT signalling using western blot and immunofluorescence staining assays. Reduced CDH11 expression in the TNBC cells altered their morphology from spindle-like to polygonal, reduced cell viability, and increased cell doubling time (Figure 3A). Additionally, our western blot data showed that MDA-MB-231 and Hs578t cells transfected with siCDH11 exhibited significantly decreased expression levels of CDH11, β-catenin, Met, c-Myc, c-Jun, and MMP7 (Figures 3B-D). Furthermore, we observed that siCDH11 in MDA-MD-231 and Hs578t cells reduced β-catenin nuclear localization and expression (Figure 4). These data corroborate the modulatory role of CDH11 on the Wnt/β-catenin signalling pathway and suggest that CDH11 plays an important role in the maintenance of β-catenin protein in the MDA-MB-231 and Hs578t TNBC cell lines.

Q5: Reviewer #1:   In all the figures, text in the graphs and images needs to be bigger.     

A5: We sincerely thank the reviewer for this observation. In the revised manuscript, we have provided figures that are more representative, legible and with better resolution.

Q6: Reviewer #1:   In IHC experiment in figure 2, TNBC patient samples should be compared to normal patient samples. Has this been done?

A6: We sincerely thank the reviewer for this suggestion. Following the reviewer’s suggestion, our revised manuscript now includes data showing the differential expression of CDH11 and b-catenin in tissues from non-tumor and TNBC subjects. Please kindly see revised Figure 2 and its legend, Page 11, Lines 308 – 314.

Figure 2. CDH11 positively modulates β-catenin expression and activates the canonical WNT signalling pathway. Statistical analyses of the TNBC sub-group of the METABRIC Breast Cancer cohort (n = 1904) show positive correlation between expression of CDH11 and (A) β-catenin, (B) WNT2, and (C) TCF2. (D and E) Differential expression of CDH11 and β-catenin in non-tumor and TNBC tissue samples demonstrated by representative IHC images and percentages of patients. Scale bar: 50 μm.  (F) Histogram showing positive correlation between CDH11 and β-catenin expression level in our TNBC cohort (n = 38).

Also see Results section, Pages 10- 11, Lines 285-312.

CDH11 positively modulates β-catenin expression and activates the canonical WNT signalling pathway

β-catenin is a key component of the canonical WNT signalling pathway, and has been shown to directly bind to the cytoplasmic domain of CDH11 (13). We hypothesized that CDH11 plays an important role in enhancing the stemness, migration, and invasion potential of TNBC cells through activation of the canonical WNT signalling pathway. Using the METABRIC-Breast Cancer (IlluminaHiSeq) cohort dataset (n = 1904), we investigated the correlation between CDH11 and several components of the WNT signalling pathway in the TNBC subgroup. We found that CDH11 expression positively correlated with β-catenin, WNT2, and TCF2 expression with pearson’s r = 0.29 (p = 8.3e-30), 0.60 (p = 7.8e-183) and 0.48 (p = 1.3e-101), respectively (Figure 2A-C). Corroborating the results above, our IHC data demonstrated that tissues from TNBC patients concurrently overexpressed CDH11 and β-catenin proteins, compared to the weak/non-expression in non-tumor tissues (Figure 2D-F). In fact, correlative analysis of our local TNBC cohort (n = 38) indicate a strong positive correlation between CDH11 and b-catenin (r = 0.62; p<0.0001; Figure 2F, lower panel). To reconfirm these findings and establish the functional significance of CDH11 expression and/or activity in TNBC cells, we knocked-down CDH11 in MDA-MB-231 and Hs578t TNBC cell lines and examined its effect on WNT signalling using western blot and immunofluorescence staining assays. Reduced CDH11 expression in the TNBC cells altered their morphology from spindle-like to polygonal, reduced cell viability, and increased cell doubling time (Figure 3A-B). Additionally, our western blot data showed that MDA-MB-231 and Hs578t cells transfected with siCDH11 exhibited significantly decreased expression levels of CDH11, β-catenin, Met, c-Myc, c-Jun, and MMP7 (Figures 3C-D). Furthermore, we observed that siCDH11 in MDA-MD-231 and Hs578t cells reduced β-catenin nuclear localization and expression (Figure 4). These data corroborate the modulatory role of CDH11 on the Wnt/β-catenin signalling pathway and suggest that CDH11 plays an important role in the maintenance of β-catenin protein in the MDA-MB-231 and Hs578t TNBC cell lines.

Q7: Reviewer #1:   Line 298 should state that Cdh11 plays an important role in the maintenance of beta-catenin protein on the surface in MDA-MB-231 and Hs578t cell lines.

A7: We appreciate the reviewer’s flair for specificity. We have now reworded the sentence to reflect the reviewer’s concern. Please kindly see revised Figure 2 and its legend, Page 11, Lines 308 – 314.

CDH11 positively modulates β-catenin expression and activates the canonical WNT signalling pathway

β-catenin is a key component of the canonical WNT signalling pathway, and has been shown to directly bind to the cytoplasmic domain of CDH11 (13). We hypothesized that CDH11 plays an important role in enhancing the stemness, migration, and invasion potential of TNBC cells through activation of the canonical WNT signalling pathway. Using the METABRIC-Breast Cancer (IlluminaHiSeq) cohort dataset (n = 1904), we investigated the correlation between CDH11 and several components of the WNT signalling pathway in the TNBC subgroup. We found that CDH11 expression positively correlated with β-catenin, WNT2, and TCF2 expression with pearson’s r = 0.29 (p = 8.3e-30), 0.60 (p = 7.8e-183) and 0.48 (p = 1.3e-101), respectively (Figure 2A-C). Corroborating the results above, our IHC data demonstrated that tissues from TNBC patients concurrently overexpressed CDH11 and β-catenin proteins, compared to the weak/non-expression in non-tumor tissues (Figure 2D-F). In fact, correlative analysis of our local TNBC cohort (n = 38) indicate a strong positive correlation between CDH11 and b-catenin (r = 0.62; p<0.0001; Figure 2F, lower panel). To reconfirm these findings and establish the functional significance of CDH11 expression and/or activity in TNBC cells, we knocked-down CDH11 in MDA-MB-231 and Hs578t TNBC cell lines and examined its effect on WNT signalling using western blot and immunofluorescence staining assays. Reduced CDH11 expression in the TNBC cells altered their morphology from spindle-like to polygonal, reduced cell viability, and increased cell doubling time (Figure 3A-B). Additionally, our western blot data showed that MDA-MB-231 and Hs578t cells transfected with siCDH11 exhibited significantly decreased expression levels of CDH11, β-catenin, Met, c-Myc, c-Jun, and MMP7 (Figures 3C-D). Furthermore, we observed that siCDH11 in MDA-MD-231 and Hs578t cells reduced β-catenin nuclear localization and expression (Figure 4). These data corroborate the modulatory role of CDH11 on the Wnt/β-catenin signalling pathway and suggest that CDH11 plays an important role in the maintenance of β-catenin protein in the MDA-MB-231 and Hs578t TNBC cell lines.

Q8: Reviewer #1:   In figure 5F, only Klf4 and c-Myc are significantly decreased from the blots. Sox2 is only decreased in the MDA cell line. Cd44 levels do not look decreased. The authors need to correct their interpretation of the data.

A8: We thank the reviewer for this careful observation. We have now provided more representative data in our revised manuscript. Please kindly see our revised Figure 5 and legend, Page 15, Lines 359 – 369.

Figure 5. CDH11 inhibition suppresses the stem cell-like phenotype of TNBC cells. (A) Images showing tumorsphere size in wild-type cells and their siCDH11 counterparts of varying cell density. Histograms comparing the tumorsphere sizes of wild-type or si-CDH11 (B) MDA-MB-231 and (C) Hs578t.  Histograms comparing the number of tumorspheres formed by wild-type or si-CDH11 (D) MDA-MB-231 and (E) Hs578t cells, based on imageJ quantification. Only tumorspheres ≥40 μm in size were counted. (F) The effect of siCDH11 on the expression Sox2, KLF4, CD44, and c-Myc proteins in MDA-MB-231 and Hs578t cells. (G) Graphical representation of the migration (upper panel) and invasion (lower panel) of MDA-MB-231 and Hs578t cells. Results represent mean ± SD of 3 independent assays in triplicate. *p<0.05, **p<0.01, ***p<0.001; WT, wild type; NS, not significant; Scale bar: 100 μm. 

Q9: Reviewer #1:   In lines 341-343, the authors use TNBC cells. This should be changed to the specific cell lines they have used (MDA and hs578t) or they should state TNBC cell lines.

A9: Once again, we appreciate the reviewer’s flair for specificity. We have made use of the reviewer’s suggestion. Kindly see our revised Results section, Page 14, Lines 355 – 357.

These data demonstrate that tumorsphere formation efficiency is suppressed in the CDH11-deficient MDA-MB-231 and Hs578t TNBC cell lines and suggest a role for CDH11 expression in the modulation of the stem cell-like phenotype of TNBC cells.

Q10: Reviewer #1:   In line 444, expression should be changed to levels.

A10: We thank the reviewer for this comment. As suggested, we have made use of ‘levels’. Kindly see our revised Discussion section, Page 19, Lines 457 – 462.

Validating our hypothesis that CDH11 may promote β-catenin nuclear translocation and WNT activation, we demonstrated that CDH11 modulates β-catenin expression levels and that CDH11/β-catenin signalling axis plays a regulatory role for the canonical WNT signalling pathway in TNBC (Figure 3), as concomitant decreased CDH11 and β-catenin in the TNBC cell was observed, with siCDH11 cells exhibiting reduced CDH11/β-catenin nuclear co-localization compared to the control WT cells (Figure 4).

Reviewer 2 Report

In the present study Satriyo et al. show that the cadherin family member CDH11 could have a tumour promoting role in triple negative breast cancer. They find a correlation between high expression of CDH11 and poor prognosis by gene expression analysis of the TCGA breast cancer cohort. Immunohistochemistry analysis of a cohort of 38 TNBC tumours showed that CDH11 high expressing cells frequently display high beta-catenin expression. Next, by using siRNA to reduce CDH11 expression in two TNB cell lines and performing a set of in vitro and in vivo studies, the authors propose a role for CDH11 in the regulation of the Wnt pathway and the CSC properties in TNBC. While the findings described in this manuscript are interesting my main concern is that often the affirmations made by the authors are not supported by their data.

 Major concerns:

They authors show that there is a correlation between high expression of CDH11 and poor prognosis in the TNBC (and basal-like) tumours of the TCGA cohort. What is the threshold of CDH11 expression that was used to discriminate “Low CDH11” and “High CDH11” tumours?

Regarding the results of the survival analysis of the TCGA cohort tumours, the authors write (Figure 1, line 269) “Patients with basal-like and TNBC are characterized by high expression of CDH11 and worse prognosis.” The claim is wrong and should be changed. Patients with BL and TNBC tumours are not characterised by CDH11 high expression. In fact, high CDH11 expression is more frequent in Luminal A and B tumours (210 out of 228 and 115 out of 123, respectively) than in BL or TNBC tumours (34 out of 96 and 96 out of 123).

Based on the correlation between CDH11 expression with beta-catenin, WNT2 and TCF2 expression levels in the TNBC cohort of the TCGA, the authors affirm that “CDH11 positively modulates β-catenin expression and activates the canonical WNT signalling pathway” (lines 276-277 and 301-302). A correlation does not demonstrate a case-effect relationship. In addition, showing the Pearson’s r for the correlation is insufficient. This parameter indicates the strength and the sign of the correlation but not its statistical significance.

A complete correlation analysis (strength, sign and significance) for CDH11 and beta-catenin staining is missing for the 38 tumours of the immunohistochemistry cohort.

In lines 290-292 the authors state that “Reduced CDH11 expression in the TNBC cells altered their morphology from spindle-like to polygonal, reduced cell viability, and increased cell doubling time (Figure 3A).” However, they only show data for the changes in the morphology of the cells. They DO NOT show any experiments studying cell viability or doubling times. Either they show the experiments or they clearly indicate in the text that these data are not shown.

SiRNA experiments should have been done using a non-targeting siRNA as a negative control instead of (or in addition to) wild type cells.

Regarding the capacity of CDH11 inhibition to suppress the stem cell-like phenotype of TNBC, the affirmation that “siCDH11 cells displayed lesser tumorsphere formation efficiency as reflected by … fewer formed tumorspheres compared to the WT cells” (lines 331-333) is not supported by the data. The number of tumour spheres formed did not differ between MDA-MB-231 WT and siCDH11 cells, and in the Hs578t cells it was different at two seeding concentrations but not in another.

It is surprising that in the xenograft studies a short-term CDH11 silencing (siRNA) achieves long-term consequences in tumour growth. The authors should work on an explanation for this.

The graphical abstract is just a suggestion as to how the system could be working, the affirmation that “CDH11 inhibition suppresses the TNBC SC-like and metastatic phenotypes through WNT signalling regulation” (lines 395-396)

Importantly, in the results and discussion sections the text should be changed to reflect the abovementioned concerns.

Author Response

Q1: Reviewer #2:   In the present study Satriyo et al. show that the cadherin family member CDH11 could have a tumour promoting role in triple negative breast cancer. They find a correlation between high expression of CDH11 and poor prognosis by gene expression analysis of the TCGA breast cancer cohort. Immunohistochemistry analysis of a cohort of 38 TNBC tumours showed that CDH11 high expressing cells frequently display high beta-catenin expression. Next, by using siRNA to reduce CDH11 expression in two TNB cell lines and performing a set of in vitro and in vivo studies, the authors propose a role for CDH11 in the regulation of the Wnt pathway and the CSC properties in TNBC. While the findings described in this manuscript are interesting my main concern is that often the affirmations made by the authors are not supported by their data.

A1: We sincerely thank the reviewer for the time taken to review our work and the important suggestions given.

Q2: Reviewer #2: They authors show that there is a correlation between high expression of CDH11 and poor prognosis in the TNBC (and basal-like) tumours of the TCGA cohort. What is the threshold of CDH11 expression that was used to discriminate “Low CDH11” and “High CDH11” tumours?

A2: We appreciate the reviewer comment. For our low/high expression group dichotomization, we did not use the traditional median or mean cutoff values, rather we employed a bioinformatics approach using the automated ‘Kaplan scan’ cutoff function of the R2 genomic interface platform (https://hgserver1.amc.nl/cgi-bin/r2/main.cgi). The Kaplan scan generates a Kaplan-Meier plot based on the most optimal mRNA cut-off expression level to discriminate between a good (low expression) and bad (high expression) prognosis cohort. This was followed by the Bonferroni test for statistical significance (p-value) of the KM plot.

Q3: Reviewer #2: Regarding the results of the survival analysis of the TCGA cohort tumours, the authors write (Figure 1, line 269) “Patients with basal-like and TNBC are characterized by high expression of CDH11 and worse prognosis.” The claim is wrong and should be changed. Patients with BL and TNBC tumours are not characterised by CDH11 high expression. In fact, high CDH11 expression is more frequent in Luminal A and B tumours (210 out of 228 and 115 out of 123, respectively) than in BL or TNBC tumours (34 out of 96 and 96 out of 123).

A3: We appreciate the reviewer’s comment. We have now reworded the figure legend to address the reviewer’s concern. Please kindly see our revised Figure 1 legend, Page 10, Lines 276 – 281.

Figure 1. Patients with TNBC and basal-like breast cancer exhibiting high expression of CDH11 are characterized by worse prognosis. Kaplan-Meier plots based on analyses of breast cancer cohort (n = 1247) using the TCGA Breast Cancer (BRCA) cohort datasets show the effect of CDH11 expression on the overall survival rates in patients regardless of molecular subtype, (A) Luminal A, (B) Luminal B, (C) Her2-enriched, (D) Basal-like, or (e) Triple negative breast cancer.

Q4: Reviewer #2: Based on the correlation between CDH11 expression with beta-catenin, WNT2 and TCF2 expression levels in the TNBC cohort of the TCGA, the authors affirm that “CDH11 positively modulates β-catenin expression and activates the canonical WNT signalling pathway” (lines 276-277 and 301-302). A correlation does not demonstrate a case-effect relationship. In addition, showing the Pearson’s r for the correlation is insufficient. This parameter indicates the strength and the sign of the correlation but not its statistical significance. A complete correlation analysis (strength, sign and significance) for CDH11 and beta-catenin staining is missing for the 38 tumours of the immunohistochemistry cohort.

A4: We are very grateful to the reviewer for very important suggestion. While we have not alluded causal effect for CDH11 expression in TNBC, we affirm that CDH11 positively modulates b-catenin expression and activates the canonical WNT signaling pathway. To address the reviewer’s concern, we have more representation data regarding same. Added to the Pearson’s r indicating strength of correlation, we have also included the p-value showing robustness of data and statistical significance. Please kindly see our new Figure 2 and its legend, Page 11, Lines 315 – 321.  

Figure 2. CDH11 positively modulates β-catenin expression and activates the canonical WNT signalling pathway. Statistical analyses of the TNBC sub-group of the METABRIC Breast Cancer cohort (n = 1904) show positive correlation between expression of CDH11 and (A) β-catenin, (B) WNT2, and (C) TCF2. (D and E) Differential expression of CDH11 and β-catenin in non-tumor and TNBC tissue samples demonstrated by representative IHC images and percentages of patients. Scale bar: 50 μm.  (F) Histogram showing positive correlation between CDH11 and β-catenin expression level in our TNBC cohort (n = 38).

Also, kindly see our revised Results section, Pages 10-11, Lines 285-312.

CDH11 positively modulates β-catenin expression and activates the canonical WNT signalling pathway

β-catenin is a key component of the canonical WNT signalling pathway, and has been shown to directly bind to the cytoplasmic domain of CDH11 (13). We hypothesized that CDH11 plays an important role in enhancing the stemness, migration, and invasion potential of TNBC cells through activation of the canonical WNT signalling pathway. Using the METABRIC-Breast Cancer (IlluminaHiSeq) cohort dataset (n = 1904), we investigated the correlation between CDH11 and several components of the WNT signalling pathway in the TNBC subgroup. We found that CDH11 expression positively correlated with β-catenin, WNT2, and TCF2 expression with pearson’s r = 0.29 (p = 8.3e-30), 0.60 (p = 7.8e-183) and 0.48 (p = 1.3e-101), respectively (Figure 2A-C). Corroborating the results above, our IHC data demonstrated that tissues from TNBC patients concurrently overexpressed CDH11 and β-catenin proteins, compared to the weak/non-expression in non-tumor tissues (Figure 2D-F). In fact, correlative analysis of our local TNBC cohort (n = 38) indicate a strong positive correlation between CDH11 and b-catenin (r = 0.62; p<0.0001; Figure 2F, lower panel). To reconfirm these findings and establish the functional significance of CDH11 expression and/or activity in TNBC cells, we knocked-down CDH11 in MDA-MB-231 and Hs578t TNBC cell lines and examined its effect on WNT signalling using western blot and immunofluorescence staining assays. Reduced CDH11 expression in the TNBC cells altered their morphology from spindle-like to polygonal, reduced cell viability, and increased cell doubling time (Figure 3A-B). Additionally, our western blot data showed that MDA-MB-231 and Hs578t cells transfected with siCDH11 exhibited significantly decreased expression levels of CDH11, β-catenin, Met, c-Myc, c-Jun, and MMP7 (Figures 3C-D). Furthermore, we observed that siCDH11 in MDA-MD-231 and Hs578t cells reduced β-catenin nuclear localization and expression (Figure 4). These data corroborate the modulatory role of CDH11 on the Wnt/β-catenin signalling pathway and suggest that CDH11 plays an important role in the maintenance of β-catenin protein in the MDA-MB-231 and Hs578t TNBC cell lines.

Q5: Reviewer #2: In lines 290-292 the authors state that “Reduced CDH11 expression in the TNBC cells altered their morphology from spindle-like to polygonal, reduced cell viability, and increased cell doubling time (Figure 3A).” However, they only show data for the changes in the morphology of the cells. They DO NOT show any experiments studying cell viability or doubling times. Either they show the experiments or they clearly indicate in the text that these data are not shown.

A5: We thank the reviewer for this comment. We have included data depicting cell viability and doubling time in our revised manuscript. Please kindly see our new Figure 3 and its legend, Page 13, Lines 323 – 331. 

Figure 3. Loss of CDH11 function using siRNA changes cell morphology and inhibits activation of the canonical Wnt signalling pathway in TNBC cells. (A) The cell morphology of MDA-MB-231 and Hs578t changed from spindle-like to polygonal after loss of CDH11 expression. (B) Graphical representation of the effect of siCDH11 on the doubling time of MDA-MB-231 and Hs578t cells ad determined by trypan-blue exclusion. (C) Western blot results show CDH11 knockdown decreased β-catenin, Met, c-Myc, c-Jun, and MMP7 protein expression level in MDA-MB-231 and Hs578t cell lines. (D) Graphical representation of C. Results expressed as mean ± SD of assays done 3 times in triplicate. *p<0.05, **p<0.01, ***p<0.001; NS, not significant; Red arrow, floating non-viable cells.

Q6: Reviewer #2: Regarding the capacity of CDH11 inhibition to suppress the stem cell-like phenotype of TNBC, the affirmation that “siCDH11 cells displayed lesser tumorsphere formation efficiency as reflected by … fewer formed tumorspheres compared to the WT cells” (lines 331-333) is not supported by the data. The number of tumour spheres formed did not differ between MDA-MB-231 WT and siCDH11 cells, and in the Hs578t cells it was different at two seeding concentrations but not in another.

A6: We thank the reviewer for this important observation. We apologize for the data mix-up and have now provided more representative data in the revised manuscript. Please kindly see our revised Figure 5 and legend, Page 15, Lines 359 – 369.

Figure 5. CDH11 inhibition suppresses the stem cell-like phenotype of TNBC cells. (A) Images showing tumorsphere size in wild-type cells and their siCDH11 counterparts of varying cell density. Histograms comparing the tumorsphere sizes of wild-type or si-CDH11 (B) MDA-MB-231 and (C) Hs578t.  Histograms comparing the number of tumorspheres formed by wild-type or si-CDH11 (D) MDA-MB-231 and (E) Hs578t cells, based on imageJ quantification. Only tumorspheres ≥40 μm in size were counted. (F) The effect of siCDH11 on the expression Sox2, KLF4, CD44, and c-Myc proteins in MDA-MB-231 and Hs578t cells. (G) Graphical representation of the migration (upper panel) and invasion (lower panel) of MDA-MB-231 and Hs578t cells. Results represent mean ± SD of 3 independent assays in triplicate. *p<0.05, **p<0.01, ***p<0.001; WT, wild type; NS, not significant; Scale bar: 100 μm. 

Also, kindly see our revised Results section, Page 14-15, Lines 342 – 357.

CDH11 inhibition suppresses the stem cell-like phenotype of TNBC cells

CSCs constitute a subset of cancer cells with self-renewal ability and enhanced resistance to some chemotherapy agents (6). We investigated whether CDH11 plays an important role in TNBC stemness. Tumorsphere formation assays using wild-type (WT) and siCDH11 MDA-MB-231 and Hs578t cells show that the siCDH11 cells displayed lesser tumorsphere formation efficiency compared to the WT cells (Figure 5A-E). Measurement of tumorsphere sizes showed that all the siCDH11 tumorspheres were significantly smaller than those formed by their WT counterparts of both cell lines except in the lowest seeding cells of Hs578t (Figure 5B and C). The siCDH11 cells produced fewer tumorspheres than WT cells at all seeding density, except 4000 cells/cm2 (Figure 5D and E).  To further confirm the important role of CDH11 in TNBC stemness, we assessed several CSC markers. We observed significantly downregulated Sox2, KLF4, CD44, and c-Myc protein expression levels in the siCDH11 group compared to the WT group (Figure 5F and Supplementary Figure 1A). These data demonstrate that tumorsphere formation efficiency is suppressed in the CDH11-deficient MDA-MB-231 and Hs578t TNBC cell lines and suggest a role for CDH11 expression in the modulation of the stem cell-like phenotype of TNBC cells.

Q7: Reviewer #2: It is surprising that in the xenograft studies a short-term CDH11 silencing (siRNA) achieves long-term consequences in tumour growth. The authors should work on an explanation for this.

A7: We appreciate the reviewer’s comment and do understand the rationale for the reviewer’s surprise, as we would have been too considering that shRNA and siRNA are transient RNAi methods. We have now expatiated more on how RNAi effect was maintained over the course of the in vivo assay in the siCDH11 murine group. We apologize for this oversight. Please kindly refer to our revised Materials & Methods section, Page 8, Lines 224-240.

In vivo study

The MDA-MB-231 and Hs578t cells were treated with siRNA specific for CDH11 for 48h. NOD/SCID mice were randomized into wild type (n=5) and siCDH11 (n=5) group for each cell line. The mice were inoculated subcutaneously with 2 x 106 wild type (WT) or siCDH11 MDA-MB-231 or Hs578t cells in their hind flank. Mice tumor sizes were measured on days 6, 9, 12, 15, 18, and 27 after TNBC cell inoculation using callipers and tumor volumes calculated with a standard formula: length x width2 x 0.5. The tumor-bearing mice were sacrificed, and the tumor mass were observed and measured on day 27 post-inoculation. In parallel in vivo studies following same steps except sacrificing on day 27, the mice were observed until day 45 post-inoculation to assess the effect of altered CDH11 expression on the survival rates.  For maintenance of siRNA effect, mice in the siCDH11 group were injected intra-tumorally with 10 μmol/L of ‘siCDH11-atelocollagen’ complex on days 9, 18, 27, and 36 after TNBC cell inoculation. The siCDH-atelocollagen complex, usually prepared the preceding evening and stored at 4oC before use, consisted of equal volumes of well mixed siCDH11 and atelocollagen in PBS at pH 7.4 (Koken Co. Ltd., Tokyo, Japan). All tumor xenograft animal studies were performed in accordance with protocol approved by the Institutional Animal Care and Use Committee.

Q8: Reviewer #2: The graphical abstract is just a suggestion as to how the system could be working, the affirmation that “CDH11 inhibition suppresses the TNBC SC-like and metastatic phenotypes through WNT signalling regulation” (lines 395-396)

A8: We thank the reviewer for the comment. We have redrawn the graphical abstract to reflect our results.  Please kindly see revised Figure 6 and legend, Page 17, Lines 402 – 417.

Figure 6. Silencing CDH11 significantly attenuates tumorigenicity of TNBC cells, in vivo. (A) Schema showing xenografts derived from NOD/SCID mice injected with 2x106 WT or siCDH11 MDA-MB-231 or Hs578t cells subcutaneously in the hind-flank and observed for 27 days (group A, n = 5/group) or 45 days (group B, n = 5/group). Graph showing difference in tumor volume over time for the WT or siCDH11 (B) MDA-MB-231 or (C) Hs578T tumor-bearing mice. Intergroup p-values were determined by 2-way ANOVA. Kaplan-Meier survival curves for mice bearing WT or siCDH11 (D) MDA-MB-231 or (E) Hs578T cell-derived tumors. WT, wild type. (F) Graphical abstract showing that CDH11 inhibition suppresses the TNBC SC-like and metastatic phenotypes through WNT signalling regulation.  In the presence of Wnt signal, cytoplasmic β-catenin is released from CDH11, accumulates in the perinuclear region, then translocates to the nucleus where it activates downstream CSCs-regulating target genes, including Met, c-Myc and Sox2 by binding to TCF/LEF. Once activated, these genes enhance self-renewal, migration, and invasion abilities of TNBC cells. In the absence of CDH11, membranar, cytoplasmic and nuclear β-catenin pool is diminished, Wnt signalling is obtused, and the canonical Wnt signalling pathway downstream target genes remains inactivated. 

Q9: Reviewer #2: Importantly, in the results and discussion sections the text should be changed to reflect the abovementioned concerns.

A9: We thank the reviewer for all the comments and suggestions made. We find them very insightful and helpful. We have revised our manuscript based on the comments and do hope we have been able to address all the reviewer’s concerns and now meet the threshold for acceptance.

Round 2

Reviewer 2 Report

Regarding my previous Q2: “What is the threshold of CDH11 expression that was used to discriminate “Low CDH11” and “High CDH11” tumours?” The answer provided by the authors should be included in the appropriate section of Materials&Methods.

Regarding Figure 2: Panels A-C of this figure evaluate the correlation between CDH11 mRNA expression levels and those of beta-catenin, WNT2 and TCF2. I have two concerns: 1) In the original version of the manuscript the expression data analysed in this figure belonged to the TCGA cohort. In this revised version of the manuscript the expression data correspond to the TNBC sub-group of the METABRIC cohort. This new cohort must be described in the Materials&Methods section. 2) In the Y axis of figure C expression data correspond to TCF12 mRNA, not to TCF2. Also, in the figure legend (line 321) “CDH11 and beta-catenin expression level” must be changed to “CDH11 and beta-catenin protein level”.

Regarding my previous Q4: The affirmation in Figure 2 “CDH11 positively modulates β-catenin expression and activates the canonical WNT signalling pathway” CANNOT be made here based on the results from this figure and MUST be changed. The fact that CDH11 mRNA expression levels positively correlate with those of beta-catenin, WNT2 and TCF2 (TCF12?), and that CDH11 and beta-catenin protein levels also correlate does not imply that this effect is directly caused by CDH11 expression.

Figure 3. The letter B for the cell count number experiments is missing in the figure. Typo in line 327 of the figure legend “ad determined” should be “as determined”.

Regarding my previous Q6: I am astonished of how the data in Figure 5D-E regarding the size of the tumorspheres from not being significant in the initial version of the manuscript to significant in the reviewed version.

Regarding my previous Q7: I am surprised that the authors forgot to include in the In Vivo Study section of Materials & Methods the fact that the animals received several injections of siCDH11 atelocollagen complex. Can the authors provide a WB for these tumours at days 15-25 showing reduced CDH11 levels in siCDH11 tumours as compared to WT?

Author Response

Answers to the comments:

Point-by-point responses to reviewer’s comments:

We would like to thank the reviewer for the thorough reading of our manuscript as well as their valuable comments. We have followed their comments closely and feel that they have further strengthened the manuscript. Below are our point-by-point responses.

Q1: Reviewer #1: Regarding my previous Q2: “What is the threshold of CDH11 expression that was used to discriminate “Low CDH11” and “High CDH11” tumours?” The answer provided by the authors should be included in the appropriate section of Materials&Methods.

A1: As with the previous review, we appreciate the reviewer comment. In this R2 version of our manuscript, we have included the answer provided earlier in appropriate section of Materials & Methods. Please kindly see our R2 Revised Materials & Methods section, Page 4-5, Lines 91-113.

Public dataset analyses

The TCGA Breast Cancer (BRCA) cohort datasets were downloaded from UCSC Xena Browser (https://xenabrowser.net/datapages/) platform. This cohort contains several datasets from 1247 samples of breast cancer patients.  We also made use of the TNBC cohort data of the Molecular Taxonomy of Breast Cancer International Consortium (METABRIC) cohort dataset (n = 1904) downloaded from the European Genome-Phenome archive (https://ega-archive.org/studies/EGAS00000000098). The METABRIC study classifies breast tumors into subcategories, based on genetic fingerprints and molecular signatures which are intended to help predict therapeutic response and determine the optimal course of treatment. The gene expression RNAseq-IlluminaHiSeq and Phenotypes datasets were downloaded and used for further analysis. The PAM50 mRNA nature2012 clinical parameter was used for classifying breast cancer patients into luminal A, luminal B, Her2-enriched and basal-like (BL) subgroups. The status of ER, PR and Her2 were used to determine the triple negative breast cancer subgroup. To establish correlation between CDH11 and prognosis of breast cancer patient for each subgroup, we performed Kaplan Meier (KM) overall survival analysis using the “R2: Genomics Analysis and Visualization Platform” (http://r2platform.com). For the low/high expression group dichotomization, we did not use the traditional median or mean cutoff values, rather we employed a bioinformatics approach using the automated ‘Kaplan scan’ cutoff function of the R2 genomic interface platform (https://hgserver1.amc.nl/cgi-bin/r2/main.cgi). The ‘Kaplan scan’ generates a KM plot based on the most optimal mRNA cut-off expression level to discriminate between a good (low expression) and bad (high expression) prognosis cohort. This was followed by the Bonferroni test for statistical significance (p-value) of the KM plot.

 Q2: Reviewer #1: Regarding Figure 2: Panels A-C of this figure evaluate the correlation between

CDH11 mRNA expression levels and those of beta-catenin, WNT2 and TCF2. I have two concerns: 1) In the original version of the manuscript the expression data analysed in this figure belonged to the TCGA cohort. In this revised version of the manuscript the expression data correspond to the TNBC subgroup of the METABRIC cohort. This new cohort must be described in the Materials&Methods section. 2) In the Y axis of figure C expression data correspond to TCF12 mRNA, not to TCF2. Also, in the figure legend (line 321) “CDH11 and beta-catenin expression level” must be changed to “CDH11 and beta-catenin protein level”.

A2: We thank the reviewer for these comments. On concern (1), as suggested by the reviewer, we have indicated that the TNBC subgroup of the METABRIC cohort was also used in the Material & Methods sections; we also provided some description of the said cohort. Please kindly see our R2 Revised Materials & Methods section, Page 4-5, Lines 91-113.

Public dataset analyses

The TCGA Breast Cancer (BRCA) cohort datasets were downloaded from UCSC Xena Browser (https://xenabrowser.net/datapages/) platform. This cohort contains several datasets from 1247 samples of breast cancer patients.  We also made use of the TNBC cohort data of the Molecular Taxonomy of Breast Cancer International Consortium (METABRIC) cohort dataset (n = 1904) downloaded from the European Genome-Phenome archive (https://ega-archive.org/studies/EGAS00000000098). The METABRIC study classifies breast tumors into subcategories, based on genetic fingerprints and molecular signatures which are intended to help predict therapeutic response and determine the optimal course of treatment. The gene expression RNAseq-IlluminaHiSeq and Phenotypes datasets were downloaded and used for further analysis. The PAM50 mRNA nature2012 clinical parameter was used for classifying breast cancer patients into luminal A, luminal B, Her2-enriched and basal-like (BL) subgroups. The status of ER, PR and Her2 were used to determine the triple negative breast cancer subgroup. To establish correlation between CDH11 and prognosis of breast cancer patient for each subgroup, we performed Kaplan Meier (KM) overall survival analysis using the “R2: Genomics Analysis and Visualization Platform” (http://r2platform.com). For the low/high expression group dichotomization, we did not use the traditional median or mean cutoff values, rather we employed a bioinformatics approach using the automated ‘Kaplan scan’ cutoff function of the R2 genomic interface platform (https://hgserver1.amc.nl/cgi-bin/r2/main.cgi). The ‘Kaplan scan’ generates a KM plot based on the most optimal mRNA cut-off expression level to discriminate between a good (low expression) and bad (high expression) prognosis cohort. This was followed by the Bonferroni test for statistical significance (p-value) of the KM plot.

Regarding the reviewer’s concern (2), we thank the reviewer for kindly pointing out this error of omission. We have now corrected these to address the reviewer’s concern. Please kindly see R2 Revised Figure 2 legend, Page 12, Lines 329-336.

Figure 2. CDH11 positively modulates β-catenin expression and activates the canonical WNT signalling pathway. Statistical analyses of the TNBC sub-group of the METABRIC Breast Cancer cohort (n = 1904) show positive correlation between expression of CDH11 and (A) β-catenin, (B) WNT2, and (C) TCF12 mRNA. (D and E) Differential expression of CDH11 and β-catenin in non-tumor and TNBC tissue samples demonstrated by representative IHC images and percentages of patients. Scale bar: 50 μm.  (F) Histogram showing positive correlation between CDH11 and β-catenin protein expression level in our TNBC cohort (n = 38).

Q3: Reviewer #1: Regarding my previous Q4: The affirmation in Figure 2 “CDH11 positively modulates β-catenin expression and activates the canonical WNT signaling pathway” CANNOT be made here based on the results from this figure and MUST be changed. The fact that CDH11 mRNA expression levels positively correlate with those of beta-catenin, WNT2 and TCF2 (TCF12?), and that CDH11 and beta-catenin protein levels also correlate does not imply that this effect is directly caused by CDH11 expression.

A3: We thank the reviewer for this comment. We have amended the subtitle and Figure 2 legend title to address the reviewer’s concern. Please kindly see our R2 Revised Results section, Page 11, Lines 299 – 326.

CDH11 positively modulates β-catenin expression and is associated with activation of the canonical WNT signalling pathway

β-catenin is a key component of the canonical WNT signalling pathway, and has been shown to directly bind to the cytoplasmic domain of CDH11 (13). We hypothesized that CDH11 plays an important role in enhancing the stemness, migration, and invasion potential of TNBC cells through activation of the canonical WNT signalling pathway. Using the METABRIC-Breast Cancer (IlluminaHiSeq) cohort dataset (n = 1904), we investigated the correlation between CDH11 and several components of the WNT signalling pathway in the TNBC subgroup. We found that CDH11 expression positively correlated with β-catenin, WNT2, and TCF2 expression with pearson’s r = 0.29 (p = 8.3e-30), 0.60 (p = 7.8e-183) and 0.48 (p = 1.3e-101), respectively (Figure 2A-C). Corroborating the results above, our IHC data demonstrated that tissues from TNBC patients concurrently overexpressed CDH11 and β-catenin proteins, compared to the weak/non-expression in non-tumor tissues (Figure 2D-F). In fact, correlative analysis of our local TNBC cohort (n = 38) indicate a strong positive correlation between CDH11 and b-catenin (r = 0.62; p<0.0001; Figure 2F, lower panel). To reconfirm these findings and establish the functional significance of CDH11 expression and/or activity in TNBC cells, we knocked-down CDH11 in MDA-MB-231 and Hs578t TNBC cell lines and examined its effect on WNT signalling using western blot and immunofluorescence staining assays. Reduced CDH11 expression in the TNBC cells altered their morphology from spindle-like to polygonal, reduced cell viability, and increased cell doubling time (Figure 3A-B). Additionally, our western blot data showed that MDA-MB-231 and Hs578t cells transfected with siCDH11 exhibited significantly decreased expression levels of CDH11, β-catenin, Met, c-Myc, c-Jun, and MMP7 (Figures 3C-D). Furthermore, we observed that siCDH11 in MDA-MD-231 and Hs578t cells reduced β-catenin nuclear localization and expression (Figure 4). These data corroborate the modulatory role of CDH11 on the Wnt/β-catenin signalling pathway and suggest that CDH11 plays an important role in the maintenance of β-catenin protein in the MDA-MB-231 and Hs578t TNBC cell lines.

Also, kindly see our R2 Revised Figure 2 legend, Page 12, Lines 329-336.

Figure 2. CDH11 positively correlates with β-catenin expression and is associated with activation of effector components of the canonical WNT signalling pathway. Statistical analyses of the TNBC sub-group of the METABRIC Breast Cancer cohort (n = 1904) show positive correlation between expression of CDH11 and (A) β-catenin, (B) WNT2, and (C) TCF12 mRNA. (D and E) Differential expression of CDH11 and β-catenin in non-tumor and TNBC tissue samples demonstrated by representative IHC images and percentages of patients. Scale bar: 50 μm.  (F) Histogram showing positive correlation between CDH11 and β-catenin protein expression level in our TNBC cohort (n = 38).

Q4: Reviewer #1: Figure 3. The letter B for the cell count number experiments is missing in the

figure. Typo in line 327 of the figure legend “ad determined” should be “as determined”.

A4: We thank the reviewer for this important observation. We have now corrected these errors. Please kindly see our R2 Revised Figure 3 legend, Page 13, Lines 338 – 346.

Figure 3. Loss of CDH11 function using siRNA changes cell morphology and inhibits activation of the canonical Wnt signalling pathway in TNBC cells. (A) The cell morphology of MDA-MB-231 and Hs578t changed from spindle-like to polygonal after loss of CDH11 expression. (B) Graphical representation of the effect of siCDH11 on the doubling time of MDA-MB-231 and Hs578t cells as determined by trypan-blue exclusion. (C) Western blot results show CDH11 knockdown decreased β-catenin, Met, c-Myc, c-Jun, and MMP7 protein expression level in MDA-MB-231 and Hs578t cell lines. (D) Graphical representation of C. Results expressed as mean ± SD of assays done 3 times in triplicate. *p<0.05, **p<0.01, ***p<0.001; NS, not significant; Red arrow, floating non-viable cells.

 Q5: Reviewer #1: Regarding my previous Q6: I am astonished of how the data in Figure 5D-E regarding the size of the tumorspheres from not being significant in the initial version of the manuscript to significant in the reviewed version.

A6: We appreciate the reviewer astonishment and do apologize for same. Please kindly see our R2 Revised Results section, Page 14-15, Lines 357 – 372.

CDH11 inhibition suppresses the stem cell-like phenotype of TNBC cells

CSCs constitute a subset of cancer cells with self-renewal ability and enhanced resistance to some chemotherapy agents (6). We investigated whether CDH11 plays an important role in TNBC stemness. Tumorsphere formation assays using wild-type (WT) and siCDH11 MDA-MB-231 and Hs578t cells show that the siCDH11 cells displayed lesser tumorsphere formation efficiency compared to the WT cells (Figure 5A-E). Measurement of tumorsphere sizes showed that all the siCDH11 tumorspheres were significantly smaller than those formed by their WT counterparts of both cell lines except in the lowest seeding cells of Hs578t (Figure 5B and C). The siCDH11 cells produced fewer tumorspheres than WT cells at all seeding density, except 4000 cells/cm2 (Figure 5D and E).  To further confirm the important role of CDH11 in TNBC stemness, we assessed several CSC markers. We observed significantly downregulated Sox2, KLF4, CD44, and c-Myc protein expression levels in the siCDH11 group compared to the WT group (Figure 5F and Supplementary Figure 1A). These data demonstrate that tumorsphere formation efficiency is suppressed in the CDH11-deficient MDA-MB-231 and Hs578t TNBC cell lines and suggest a role for CDH11 expression in the modulation of the stem cell-like phenotype of TNBC cells.

Also, kindly see our R2 Revised Figure 5 legend, Page 15-16, Lines 374 – 384.

Figure 5. CDH11 inhibition suppresses the stem cell-like phenotype of TNBC cells. (A) Images showing tumorsphere size in wild-type cells and their siCDH11 counterparts of varying cell density. Histograms comparing the tumorsphere sizes of wild-type or si-CDH11 (B) MDA-MB-231 and (C) Hs578t.  Histograms comparing the number of tumorspheres formed by wild-type or si-CDH11 (D) MDA-MB-231 and (E) Hs578t cells, based on ImageJ quantification. Only tumor spheres ≥40 μm in size were counted. (F) The effect of siCDH11 on the expression Sox2, KLF4, CD44, and c-Myc proteins in MDA-MB-231 and Hs578t cells. (G) Graphical representation of the migration (upper panel) and invasion (lower panel) of MDA-MB-231 and Hs578t cells. Results represent mean ± SD of 3 independent assays in triplicate. *p<0.05, **p<0.01, ***p<0.001; WT, wild type; NS, not significant; Scale bar: 100 μm.

Q6: Reviewer #1: Regarding my previous Q7: I am surprised that the authors forgot to include in the In Vivo Study section of Materials & Methods the fact that the animals received several injections of siCDH11 atelocollagen complex. Can the authors provide a WB for these tumours at days 15-25 showing reduced CDH11 levels in siCDH11 tumours as compared to WT?

 A6: We appreciate the reviewer astonishment. We would have been as well; however, this was not intended. Having said that, we are grateful to the reviewer for helping us make sure that every bit of our experimental procedure was carefully and conscientiously included in the manuscript, thus, helping to preclude ambiguity and prevent the assumption that well-informed reader would automatically understand what is not penned down. We thank the reviewer and do apologize for this oversight once again. Please kindly refer to our R2 Revised Materials & Methods section, Page 8-9, Lines 236-253.

In vivo study

The MDA-MB-231 and Hs578t cells were treated with siRNA specific for CDH11 for 48h. NOD/SCID mice were randomized into wild type (n=5) and siCDH11 (n=5) group for each cell line. The mice were inoculated subcutaneously with 2 x 106 wild type (WT) or siCDH11 MDA-MB-231 or Hs578t cells in their hind flank. Mice tumor sizes were measured on days 6, 9, 12, 15, 18, and 27 after TNBC cell inoculation using callipers and tumor volumes calculated with a standard formula: length x width2 x 0.5. The tumor-bearing mice were sacrificed, and the tumor mass were observed and measured on day 27 post-inoculation. In parallel in vivo studies following same steps except sacrificing on day 27, the mice were observed until day 45 post-inoculation to assess the effect of altered CDH11 expression on the survival rates.  For maintenance of siRNA effect, mice in the siCDH11 group were injected intra-tumorally with 10 μmol/L of ‘siCDH11-atelocollagen’ complex on days 9, 18, 27, and 36 after TNBC cell inoculation. The siCDH-atelocollagen complex, usually prepared the preceding evening and stored at 4oC before use, consisted of equal volumes of well mixed siCDH11 and atelocollagen (Koken Co. Ltd., Tokyo, Japan). All tumor xenograft animal studies were approved by the TMU-SHH Joint Institutional Review Board and performed in accordance with protocol approved by the TMU Institutional Animal Care and Use Committee.

We thank the reviewer for all the comments and suggestions made. We find them non-prejudicial and helpful. We have revised our manuscript once again based on these comments and do hope we have now addressed all the reviewer’s concerns and now meet the threshold for acceptance.
